

# Antarctic climate variability at regional and continental scales over the last 2,000 years

Barbara Stenni[1,2], Mark A. J. Curran[3,4], Nerilie J. Abram[5,6], Anais Orsi[7], Sentia Goursaud[7,8], Valerie Masson-Delmotte[7], Raphael Neukom[9], Hugues Goosse[10], Dmitry Divine[11,12], Tas van Ommen[3,4], Eric J. Steig[13], Daniel A. Dixon[14], Elizabeth R. Thomas[15], Nancy A. N. Bertler[16,17], Elisabeth Isaksson[11], Alexey Ekaykin[18,19], Massimo Frezzotti[20], Martin Werner[21]

[1]Department of Environmental Sciences, Informatics and Statistics, Ca' Foscari University of Venice, Italy

[2]Institute for the Dynamics of Environmental Processes, CNR, Venice, Italy

[3]Australian Antarctic Division, 203 Channel Highway, Kingston Tasmania 7050, Australia

[4]Antarctic Climate & Ecosystems Cooperative Research Centre, University of Tasmania, Hobart 7001, Australia

[5]Research School of Earth Sciences, Australian National University, Canberra ACT 2601, Australia.

[6]ARC Centre of Excellence for Climate System Science, Australian National University, Canberra ACT 2601, Australia

[7]Laboratoire des Sciences du Climat et de l'Environnement (IPSL/CEA-CNRS-UVSQ UMR 8212), CEA Saclay, 91191 Gif-sur-Yvette cédex, France

[8]Université Grenoble Alpes, Laboratoire de Glaciologie et Géophysique de l'Environnement (LGGE), 38041 Grenoble, France

[9]University of Bern, Oeschger Centre for Climate Change Research & Institute of Geography, 3012 Bern, Switzerland

[10]Université catholique de Louvain, Earth and Life Institute, Centre de recherches sur la terre et le climat Georges Lemaître, B-1348 Louvain-la-Neuve, Belgium

[11]Norwegian Polar Institute, Fram centre, N-9296 Tromsø, Norway

[12]Department of Mathematics and Statistics, Faculty of Science, University of Tromsø - The Arctic University of Norway, N-9037, Norway

[13]Department of Earth and Space Sciences, University of Washington, Seattle, WA 98195, USA

[14]Climate Change Institute, University of Maine, Orono, ME 04469, USA

[15]British Antarctic Survey, Cambridge, UK CB3 0ET

[16]Antarctic Research Centre, Victoria University of Wellington, Wellington 6012, New Zealand

[17]National Ice Core Research Facility, GNS Science, Gracefield 5040, New Zealand

[18]Arctic and Antarctic Research Institute, St Petersburg, Russia

[19]Institute of Earth Sciences, Saint Petersburg State University, St Petersburg, Russia

[20]ENEA Casaccia, Rome, Italy

[21]Alfred Wegner Institute, Helmholtz Centre for Polar and Marine Research, 27570 Bremerhaven, Germany

*Correspondence to*: Barbara Stenni (barbara.stenni@unive.it)

**Abstract.** Climate trends in the Antarctic region remain poorly characterised, owing to the brevity and scarcity of direct climate observations and the large magnitude of interannual to decadal-scale climate variability. Here, within the framework of the PAGES Antarctica 2k working group, we build an enlarged database of ice core water stable isotope records from Antarctica, consisting of 112 records. We produce both unweighted and weighted isotopic ($\delta^{18}O$) composites and temperature reconstructions since 0 CE, binned at 5 and 10-year resolution, for 7 climatically-distinct regions covering the Antarctic continent. Following earlier work of the Antarctica 2k working group, we also produce composites and reconstructions for the broader regions of East Antarctica, West Antarctica, and the whole continent. We use three methods for our temperature reconstructions: i) a temperature scaling based on the $\delta^{18}O$-temperature relationship output from an ECHAM5-wiso model simulation nudged to ERA-interim atmospheric reanalyses from 1979 to 2013, and adjusted for the West Antarctic Ice Sheet region to borehole temperature data; ii) a temperature scaling of the isotopic normalized anomalies to the variance of the regional reanalysis temperature and iii) a composite-plus-scaling approach used in a previous continental scale reconstruction of Antarctic temperature since 1 CE but applied to the new Antarctic ice core database. Our new reconstructions confirm a significant cooling trend from 0 to 1900 CE across all Antarctic regions where records extend back into the 1st millennium,



with the exception of the Wilkes Land coast and Weddell Sea coast regions. Within this long-term cooling trend from 0-1900 CE we find that the warmest period occurs between 300 and 1000 CE, and the coldest interval from 1200 to 1900 CE. Since 1900 CE, significant warming trends are identified for the West Antarctic Ice Sheet, the Dronning Maud Land coast and the Antarctic Peninsula regions, and these trends are robust across the distribution of records that contribute to the unweighted

isotopic composites and also significant in the weighted temperature reconstructions. Only for the Antarctic Peninsula is this most recent century-scale trend unusual in the context of natural variability over the last 2000-years. However, projected warming of the Antarctic continent during the 21st Century may soon see significant and unusual warming develop across other parts of the Antarctic continent. The extended Antarctica 2k ice core isotope database developed by this working group opens up many avenues for developing a deeper understanding of the response of Antarctic climate to natural and

anthropogenic climate forcings. The first long-term quantification of regional climate in Antarctica presented herein is a basis for data-model comparison and assessments of past, present and future driving factors of Antarctic climate.

## 1 Introduction

Antarctica is the region of the world where instrumental climate records are shortest and sparsest. Estimates of temperature

change with reasonable coverage across the full Antarctic continent are only available since 1958 CE (Nicolas and Bromwich, 2014), and the large magnitude of year-to-year climate variability that characterises Antarctica makes the interpretation of trends in this data sparse region problematic (Jones et al., 2016). As a result, the knowledge of past Antarctic temperature and climate variability is predominantly dependent on proxy records from natural archives. While coastal proxy records are being developed from terrestrial and marine archives (Jones et al., 2016), information on Antarctic climate above the ice sheet

exclusively relies on the climatic interpretation of ice core records.

Within the variety of measurements performed in boreholes and ice cores, only water stable isotopes can provide subdecadal resolution records of past temperature changes (Küttel et al., 2012). In high accumulation areas of coastal zones and West Antarctica, annual layer counting is feasible during the last centuries to millennia (Plummer et al., 2012; Abram et al., 2013; Thomas et al., 2013; Sigl et al. 2016; Winstrup et al. in prep.), and annual water stable isotope signals can be delivered.

However, in the dry regions of the central Antarctic plateau, where the longest ice core records are available, chronologies are less accurate and rely on the identification of volcanic deposits that can be used to tie ice cores from different sites to a common Antarctic ice core age scale (Sigl et al., 2014 and 2015).

The chemical and physical signals measured in an individual ice core reflect a local climatic signal archived through the deposition and reworking of snow layers. The intermittency of Antarctic precipitation, variability in precipitation source

regions, and post-depositional effects of snow layers including wind drift and scouring, sublimation, and snow metamorphism can distort the climate signal preserved within ice cores and produces non-climatic noise. As a result, obtaining a robust climate signal can only be achieved through the combination of multiple ice core records from a given site and/or region, and through the site-specific calibration of the relationships between water stable isotopes and temperature.

Water can be characterised by the stable isotope ratios of oxygen ($\delta^{18}$O: the deviation of the ratio of $^{18}$O/$^{16}$O in a sample,

relative to that of the standard, Vienna Standard Mean Ocean Water) and of deuterium ($\delta$D: the deviation of the ratio of $^{2}$H/$^{1}$H). Both of these parameters within ice cores provide information on past temperatures. There is solid theoretical understanding of distillation processes relating moisture transport towards the polar regions with air mass cooling and the progressive loss of heavy water molecules along the condensation pathway (Jouzel and Merlivat, 1984). This theoretical understanding is further supported by numerical modelling performed using atmospheric general circulation models equipped with water stable

isotopes (Jouzel, 2014). The effects of these processes are observed in the spatial relationships between the isotopic composition of Antarctic precipitation/surface snow and surface air temperature across the continent. However, relationships between water stable isotopes in snow and surface temperature may vary through time as a result of changes between condensation and surface temperature (in relationship to changes in boundary layer stability), changes in moisture origin and



initial evaporation conditions, changes in atmospheric transport pathways and changes in precipitation seasonality or intermittency (Masson-Delmotte et al., 2008). Investigations based on the sampling of Antarctic precipitation have demonstrated that seasonal and inter-annual isotope versus temperature slopes are generally smaller than spatially-derived relationships (van Ommen and Morgan, 1997; Schneider et al., 2005; Stenni et al., 2016; Schlosser et al., 2004; Ekaykin et al.,

2004; Fernandoy et al., 2010). Moreover, emerging studies combining the monitoring of surface water vapour isotopic composition with the isotopic composition retained in surface snow and precipitation have revealed that snow-air isotopic exchanges during snow metamorphism affect surface snow isotopic composition (Ritter et al., 2016; Casado et al., 2016a; Casado et al., 2016b; Touzeau et al., 2016). It is not yet possible to assess the importance of such post-deposition processes for the interpretation of ice core water stable isotope records, but they may enhance the relationship between snow isotopic

composition and surface temperature more than expected from the intermittency of snowfall (Touzeau et al., 2016). Changes in ice sheet height due to ice dynamics may also affect the surface climate trends inferred from water stable isotope records; however, this influence should be of second order over the last 2000-year interval that is the focus of this study (Fegyveresi et al., 2011).

As a result, the two key challenges to reconstruct past changes in Antarctic temperature from ice core isotope records are (1)
to develop methodologies to combine different individual or stacked ice core records in order to deliver regional-scale climate signals, and (2) to quantify the temperature changes represented by water stable isotope variations.

Goosse et al. (2012) first calculated a composite of Antarctic temperature simply by averaging seven standardized temperature records inferred from water stable isotopes using a spatial isotope-temperature relationship for the last millennium. The first coordinated effort to reconstruct Antarctic temperature during the last 2000 years (PAGES 2k Consortium, 2013) screened

published ice core records for annual layer counting or alignment of volcanic sulphate records and overlap with instrumental temperature data (Steig et al., 2009), leading to the selection of 11 records. The reconstruction procedure used a composite-plus-scaling approach similar to the methodology of Schneider et al. (2006), and produced reconstructions of the continent-wide temperature history as well as specific West Antarctica and East Antarctica reconstructions. The skill of the reconstructions was limited by the number of available records through time (for instance, only one predictor in each region

prior to 166CE). This analysis identified significant (p<0.01) cooling trends from 166 to 1900 CE, 2.5 times larger in West Antarctica than in East Antarctica. A robust cooling trend over this time period has also been identified from terrestrial and marine reconstructions from other regions (PAGES 2k Consortium, 2013; McGregor et al., 2015).

The comparison of these first Antarctic 2k time series with those from other regions obtained within the PAGES 2k working groups identified three specificities: (i) Antarctic reconstructed centennial variations did not correlate with those from other

regions; (ii) the Antarctic region was the only one where a protracted cold period was not starting around 1580 CE; (iii) the Antarctic region was the only one where the 20[th] century was not the warmest century of the last 2000 years. A recent effort to characterize Antarctic and sub-Antarctic climate variability during the last 200 years also concluded that most of the trends observed since satellite climate monitoring began in 1979 CE cannot yet be distinguished from natural (unforced) climate variability (Jones et al., 2016), and are of the opposite sign to those produced by most forced climate model simulations over

the same post-1979 CE interval. The only exception to this conclusion was for changes in the Southern Annular Mode (SAM), the leading mode of atmospheric circulation variability in the high latitudes of the SH, which has showed a significant and unusual positive trend since 1979 CE.

While changes in the SAM have been related to the human influence on stratospheric ozone and greenhouse gases (Thompson et al., 2011), major gaps remain in identifying the drivers of multi-centennial Antarctic climate variability. For instance, the

influence of solar and volcanic forcing on Antarctic climate variability remains unclear. This is due to both the lack of observations and to the lack of confidence in climate model skill for the Antarctic region (Flato et al., 2013). Goosse et al. (2012) have used simulations from an intermediate complexity model to attribute the Antarctic annual mean cooling trend from 850 to 1850 CE to volcanic forcing. Recent comparisons of climate model simulations with the PAGES2k regional





reconstructions have highlighted greater model-data disagreement in the Southern Hemisphere (SH) than in the Northern Hemisphere (PAGES 2k–PMIP3 group, 2015; Abram et al 2016); such disagreement could be due either to model deficiencies or to large uncertainties in the reconstructions which were built on relatively small number of records. Changes in ocean heat content and ocean heat transport have likely contributed to the different temperature evolution at high southern latitudes

compared to other regions of the Earth (Goosse 2017), and model based studies have suggested that circulation in the Southern Ocean may act to delay by centuries the development of sustained warming trends in high southern latitudes (Armour et al., 2016). Antarctic temperature reconstructions spanning the last 2000 years may help to better constrain the processes and timescales by which natural and anthropogenic forcing act to affect climate changes in the Antarctic region.

This motivates our efforts to produce updated Antarctic temperature reconstructions. The previous continent-scale
reconstruction (PAGES 2k Consortium, 2013), where only a limited number of records have been used, may mask important regional-scale features of Antarctica's climate evolution. Here we use an expanded paleoclimate database of Antarctic ice core isotope records and new reconstruction methodologies to reconstruct the climate of the past 2000 years, at decadal scale and on a regional basis. Seven distinct climatic regions have been selected: the Antarctic Peninsula, the West Antarctic Ice Sheet, the East Antarctic Plateau, and four coastal domains of East Antarctica. This regional selection, which is supported by regional
atmospheric RACMO2.4 model results, is applied to both Antarctic ice core-derived isotopic (temperature-proxy) and snow accumulation rate reconstructions (see companion paper in the same issue by Thomas et al.). Section 2 describes the ice core and the temperature data sets used in this study, as well as the modelling framework used to support the analysis. The climate region definition, the pre-processing of the data and the different reconstruction methods are presented in Section 3. Section 4 discusses our new regional isotopic and temperature reconstructions for Antarctica, including the application of the previous
methodology to the new database. Finally, section 5 presents the summary of our results and their implications.

## 2 Datasets

### 2.1 Ice core records

Here we present and use a new expanded database which has been compiled in the framework of the PAGES Antarctica 2k
working group. The initial selection criteria are those requested by the PAGES 2k network (http://www.pages-igbp.org/ini/wg/2k-network/data) for the building of the community-sourced database of temperature-sensitive proxy records (PAGES2k Consortium, revised for Scientific Data). Briefly, i) the records must be publically available and published, ii) a relation between the climate proxies and variables should be stated, iii) the record duration should be between 300 and 2000 years, iv) the chronology, certified by the data owner, should contain at least one chronological control point near the end
(most recent) part of the record and another near the oldest part of the record, v) the resolution should be at least one analysis every 50 years.

In building the Antarctica2k database we also allow shorter records to be included, although request a stratigraphic control using volcanic markers (Sigl et al., 2014) and whenever possible, a dating by annual layer counted chronology. This last requirement is only possible in the high-accumulation regions of West Antarctica, the Antarctic Peninsula and coastal areas of
East Antarctica. The inclusion of shorter records is designed to improve data coverage for assessments of climatic trends in Antarctica during the past century. The 11 records included in the previous continental-scale reconstruction (PAGES 2k Consortium, 2013) relied on a highly precise chronological framework consisting of a common chronology, which used 42 volcanic events to synchronize the records. Here, we use both high and low-resolution records. Most of the records have a data resolution ranging from 0.025 to 5 years (only three records have a resolution of >10 years). Previous studies (Frezzotti et al.,
2007; Ekaykin et al., 2014) have shown that post-depositional and wind scouring effects, acting more effectively when the accumulation rate is very low, limit our ability to obtain temperature reconstructions at annual resolution in most of the interior of Antarctica. Because of this, in our regional reconstructions we use 5-year averaged data for reconstructing the last 200 years,



and 10-year averages for reconstructing the last 2000 years. Using 5, or 10-year averages also decreases our dependence on an annually precise chronological constraint between the ice core records, allowing us to more confidently use the expanded database. The data have been also screened for glaciological problems, with those records that are very likely to be affected by ice flow dynamics excluded.

This enlarged database consists of 112 isotopic records. A list of the records used are reported in Table S1 (Supplementary Information) and their spatial distribution is shown in Figure 1. Figure S1 shows the location of the ice core sites along with a visualization of the record lengths. Most of the records of this new database cover the last 200 years and this is particularly true for the more coastal areas. Within the database, 36 records cover just the last 50 years or less, while 50 records cover the whole length of the past 200 years. There are 15 records that cover the last 1000 years, while only 9 records reach as far back

as 0 CE.

## 2.2 Temperature product

The instrumental record is very short in Antarctica, and most ice core sites do not have weather station measurements associated with the cores. In addition, the retrieval of the first meter of firn can be difficult, due to poor cohesion of the snow. As a result, for many sites, there is no overlap between instrumental and proxy data, which complicates the proxy calibration

exercise. To enlarge the calibration dataset, we use the climate field reconstruction from Nicolas and Bromwich (2014) (hereafter NB2014). This surface temperature dataset provides homogeneous data at 60km resolution, extends from 1957 to 2013, and includes the revised Byrd temperature record (Bromwich et al., 2013) that improves the skill of the temperature product over West Antarctica. It covers a longer timespan than reliable atmospheric reanalysis products for Antarctica (which begin only 1979 CE), and has a higher spatial resolution than available isotope enabled GCM outputs. This dataset is used to

estimate the spatial representativeness of individual core sites, to scale the normalized isotopic anomaly data to temperature, and to calculate the surface temperature reconstructions with the Composite-Plus-Scale (CPS) method (Section 3.4.4).

## 2.3 Modelling framework

In order to use model information on isotope-temperature relationships in Antarctic precipitation, we use a reference simulation performed using the Atmospheric General Circulation Model ECHAM5-wiso. The initial ECHAM5 model (Roeckner et al.,

2003) has been equipped with water stable isotopes (Werner et al., 2011), following earlier work on ECHAM3 (Hoffmann et al., 1998) and ECHAM4 (Werner et al., 2001), and accounting for fractionation processes during phase changes. This model is used here because recent studies, based on model-data comparisons using observations of precipitation and surface vapour isotopic composition at a global scale and in the Arctic (e.g. Werner et al, 2011; Steen-Larsen et al., 2017), have shown strong model skill of ECHAM5-wiso when it is run in high resolution as in this study (T106, with a mean horizontal grid resolution

of approximately 1.1° x 1.1°). In Antarctica, model performance was assessed against a compilation of surface data (Masson-Delmotte et al., 2008) and recent measurements of vapour and precipitation (Goursaud et al., in preparation; Ritter et al., 2016; Dittmann et al., 2016).

Here, we use a 1958-2014 CE simulation where ECHAM5-wiso was nudged to atmospheric reanalyses from ERA40 (Uppala et al., 2005) and ERA interim (Dee et al., 2011), and run using the same ocean surface boundary conditions (SST and sea-ice)

as in ERA40 and ERA interim. Ocean surface water isotopic values were set to constant values using a compilation of observational data (Schmidt et al., 2007). Inter-comparisons of reanalysis products showed good skills of ERA-interim for Antarctic precipitation (Wang et al., 2016), surface temperature, as well as vertical profiles of winds and temperatures. However, comparisons with in-situ observations reveal an underestimate of precipitation and slight cold bias in the surface temperatures in some regions (Thomas and Bracegirdle, 2015).

The ECHAM5-wiso simulations produce a large increase in the temperature and the $\delta^{18}O$ outputs prior to 1979, which is not observed in instrumental or ice core data (Goursaud et al., 2017; Goursaud et al., in prep.). This arises from a discontinuity in



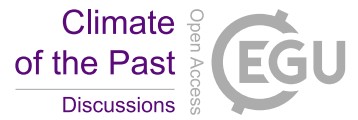

the ERA-40 reanalyses due to the lack of observations available for assimilation and boundary conditions prior to the satellite era (e.g. Antarctic sea ice) (Nicholas and Bromwich, 2014). We therefore use the ECHAM5-wiso simulations only for 1979-2013 CE.

For the analysis of the isotope-temperature relationships at each individual ice core site, we extracted the grid point data closest
to each site. For the analysis of isotope-relationships at regional scale, we calculated the area-weighted average of model outputs at grid points within the region. The $\delta^{18}O$-temperature relationship was calculated using the annual or seasonal average 2-meter temperature and annual precipitation-weighted $\delta^{18}O$, to mimic deposition processes. The simulation does not account for post-deposition processes (i.e., diffusion, which is not important on the 5 and 10-year timescales considered here; e.g. Küttel et al., 2012).

**3 Methodology**

**3.1 Defining climatic regions**

Earlier work of the PAGES Antarctica 2k working group produced a continent-scale temperature reconstruction for the whole of Antarctica, as well as reconstructions for East and West Antarctica based on a separation approximated by the Transantarctic mountain chain (PAGES 2k Consortium 2013). These broad-scale groupings mask important regional climatic trends noted in
individual studies. In particular, the absence of recent significant warming in the Antarctica 2k continent-scale temperature reconstruction is known to not be representative of all Antarctic locations (e.g. Steig et al., 2009; Mulvaney et al., 2012; Abram et al., 2013; Steig et al 2013).

In this study we choose seven climatic reconstruction regions (Figure 1). These regions are defined based on our knowledge of regional climate and snow deposition processes in the Antarctic region, as well as the availability of ice core isotope records.
The regional selections were further validated and refined by spatial correlation of temperature using the NB2014 data product. The seven climatic regions are defined as follows (see Table S1):

1. EAST ANTARCTIC PLATEAU: All East Antarctic contiguous regions at an elevation higher than 2000m, including everything south of 85°S. We exclude high peaks of the Transantarctic Mountains if they belong to the Victoria Land - Ross Sea coast (e.g. Taylor Dome or Hercules Névé).
2. WILKES LAND COAST: A region that sits at an altitude <2000m, and extends from Lambert Glacier (67°E) east to the start of Victoria Land and the Transantarctic Mountains (160°E).

3. WEDDELL SEA COAST: Extending eastward from longitude 60°W to 30°W, and south of 75°S, and lying at an altitude <2000m. Eastward of the 30°W longitude, the 75°S latitude defines the boundary with the Dronning Maud Land coast region, with the northeastern corner of the Weddell Sea coast region occurring where the 75°S latitude meets the 2000m elevation
contour. This region includes the Filchner Ice Shelf and most of the Ronne Ice Shelf.

4. ANTARCTIC PENINSULA: This region encompasses the mountainous Antarctic Peninsula. Between 74°S and 70°S the longitudinal boundaries lie between 60-80°W, while north of 70°S the longitudinal boundaries increase to 50-80°W so as to capture the northern end of the peninsula.

5. WEST ANTARCTIC ICE SHEET: A region bounded by longitudes 60°W to 170°W, and north of 85°S. In the Peninsula
region (60-80°W) a northern bound of 74°S is also applied.

6. VICTORIA LAND - ROSS SEA: This region is north of 85°S, and at an altitude <2000m, with the exception of some localised peaks within the Trans-Antarctic mountain range. It extends from 160°E to 190°E (i.e. 170°W) and incorporates most of the Ross Ice Shelf.

7. DRONNING MAUD LAND COAST: Extending eastward from 30°W to 67°E (Lambert glacier). The southern-most
boundary lies at 75°S (where this region borders with the Weddell Sea region), or at the 2000-meter elevation contour elsewhere.



In addition to these seven climatic regions, we also produce reconstructions for a continent-wide Antarctic region. Broad-scale East Antarctic (incorporating the climatic regions of the East Antarctic Plateau, as well as the Weddell Sea, Dronning Maud Land, Wilkes Land and Victoria Land coasts) and West Antarctic (incorporating the climatic regions of the West Antarctic Ice Sheet and Antarctic Peninsula) reconstructions are also presented. These additional reconstructions facilitate comparisons of

our new results, using additional methods and an expanded database, with earlier findings of the Antarctica 2k working group and subsequent research using the 2013 continent-scale reconstruction for Antarctic temperature.

### 3.2 Data pre-processing

All ice core records in the Antarctica 2k database were assigned to one of the climatic regions described in section 3.1 (as well as to East *vs.* West Antarctica, and to the Antarctic-wide classifications). The majority (94 out of 112) of ice core water isotope

records in the database are based on oxygen isotope ratios ($\delta^{18}O$). In cases where only deuterium isotope ($\delta D$) data is available, the ice core time series were converted to $\delta^{18}O$-equivalent by dividing by 8, a value that represents the slope of the global mean meteoric relationship of oxygen and deuterium isotopes in precipitation, and is close to the ratio of 7.75 observed in surface Antarctic snow (Masson-Delmotte et al., 2008).

The ice core $\delta^{18}O$ (and $\delta^{18}O$-equivalent) records were compiled on a common annual average age scale. For records with sub-

annual resolution this involved averaging all data from within a calendar year to generate an annual average dataset. Pseudo-annual records were generated for the ice core $\delta^{18}O$ records with lower than annual resolution. These pseudo annuals assume that each low-resolution isotopic value represents an average of the full time interval that the sample covers. As such, a nearest-neighbour interpolation method was used to generate stepped pseudo-annual records that continue the measured isotopic value across all of the calendar years that it represents.

Records were next binned to 5y and 10y average resolution and converted to $\delta^{18}O$ anomalies. This reduction in resolution is designed to reduce the influence of small age uncertainties between the records, as well as the non-climatic noise induced by post-deposition (e.g. wind erosion, diffusion) processes (Frezzotti et al. 2007; Ekaykin et al., 2014). The 5y resolution records were converted to anomalies relative to their mean over the 1960-1990 CE interval, and records that don't include a minimum of 6 bins (30y) coverage since 1800 CE are excluded based on length. Overall, 79 records in the new Antarctica 2k water

isotope database meet the minimum requirement of having at least 30y data coverage since 1800 CE. In some cases, records meet this minimum length requirement but do not include data for the full 1960-1990 CE reference interval. We adjust the mean value of each of these records by matching the mean $\delta^{18}O$ of their most recent 6 bins (30y) of data to the mean of all anomaly records from the same climatic region and over the same 6-bin interval. We also produce normalised records by adjusting the variance of the records using the same reference period and method as for the anomaly records.

The 10y resolution anomaly and normalised records were generated using the same method but using a reference period of 1900-1990 CE and a minimum data coverage of 9 bins (90y) since 0 CE. Similarly, records that don't include the full 1900-1990 CE reference period have the mean of their most recent 9-bin (90y) interval adjusted to match the mean of all other anomaly records from the same region and over the same 9-bin interval. Overall, 67 records in the new Antarctica 2k water isotope database meet the minimum requirement of having at least 90y data coverage since 0 CE

Some regions of Antarctica include dense networks of ice core data, such as coastal and plateau sites in Dronning Maud Land (Fig. S1). To reduce possible bias towards these data-rich regions an additional data reduction method was used, based on a 2° latitude by 10° longitude grid. Where multiple ice core records from the same climatic-region fall within the same grid cell, their isotopic anomalies (or normalised data) are averaged to produce a single composite time series for the grid. This replicates the simple unweighted compositing method described in section 3.3.1, but at the grid-scale to reduce the representation of

data-rich areas prior to the regional compositing. Supplementary figures 2 and 3 show the distribution of records by region that meet the 6-bin (30y) and 9-bin (90y) minimum requirements for the 5y and 10y composites, respectively, after the gridded data reduction step.



### 3.3 Compositing Methods

We use a suite of reconstruction methods of varying complexity in order to assess robust trends and variability in Antarctic ice core $\delta^{18}$O records and temperature.

### 3.3.1 Unweighted Composites

Our first reconstruction method involves calculating simple composites of $\delta^{18}$O anomalies. For each 5 or 10y bin we calculate the mean $\delta^{18}$O anomaly across all records in the climatic region, as well as the distribution of $\delta^{18}$O anomalies within each bin (Figures 2 and 3). This basic reconstruction method is analogous to that used for the Ocean 2k low-resolution reconstruction (McGregor et al. 2015). The benefit of this simple method is that it requires no weighting or calibration assumptions, which is advantageous for data sparse regions such as Antarctica (and the global oceans). The disadvantage is that it applies equal
weighting to all records within a climatic region, which may introduce biases related to record length, location and climatic skill.

### 3.3.2 Weighted composites based on site-level temperature regressions

In order to avoid biases from uneven data sampling, we performed, for each region, a multiple regression between each site temperature and the relevant regional average temperature (Figures 2 and 3). Most of the ice core records do not cover the full
instrumental period, so it is problematic to use the $\delta^{18}$O anomalies directly to determine the regression vectors required for a weighted temperature reconstruction. Instead, we use the climate field reconstruction of NB2014 to estimate the weights: the annual mean temperature time series at the grid cell corresponding to each ice core site is extracted from the NB2014 product, and the regional average is also calculated for each reconstruction region. Regression-based weightings are calculated based on the relationship between site temperature and regional average temperature, and the regression is performed for each
combination of ice core records through time. The weights are then applied to the ice core $\delta^{18}$O anomalies to produce regional averaged standardized anomalies.

We further reproduced this regression method using both the $\delta^{18}$O and temperature field outputs from the ECHAM5-wiso experiments. The regression of site-$\delta^{18}$O against regional average-$\delta^{18}$O, or site-temperature onto regional-temperature gave nearly identical weighting factors, supporting the use of the temperature field to calculate regressions. The effects of the
different weighting methods on each regional isotopic composite, as well as the initial 10y isotopic anomaly records, are reported in the Supplementary Information (Figures from S4 to S10). The small differences between ECHAM and NB2014-based regressions were due to the lower resolution of ECHAM-5wiso, which does not include islands and topographic features such as Roosevelt Island and Law Dome. For this reason, we preferentially use the NB2014 dataset for the temperature regression reconstruction method.

### 3.4 Temperature Reconstructions

The lack of an overlap period between our site $\delta^{18}$O records and direct temperature observations makes the proxy calibration difficult. The CPS method (Section 3.4.4), which replicates the 2013 PAGES 2k reconstruction method, is limited to sites where this calibration is possible. In order to overcome this limitation and include the largest number of records, we also use models to scale the regional isotope composites. A first method uses ECHAM5-wiso to determine the regional $\delta^{18}$O-
temperature relationship in a mechanistic way (Section 3.4.1). A second method uses a more statistical approach, and simply scales the normalized record to the instrumental period temperature variance (Section 3.4.2). Both of these approaches are equally valid and share the same hypothesis: that the instrumental period (1979-2013) is representative of the longer-term climate variability. Finally, for the West Antarctic Ice Sheet region an independent longer-term temperature record is available from borehole temperature measurements (Orsi et al., 2012). We use this independent temperature record to scale the long





term 1000-1600 temperature trend for the West Antarctic Ice Sheet region, to provide our best estimate of temperature change in line with current knowledge (Section 3.4.3).

### 3.4.1 Scaling using model-based regional δ¹⁸O-Temperature relationships

We use the coherent physical framework of the 1979-2013 CE simulation performed at T106 resolution with ECHAM5-wiso
to infer constrains on regional δ¹⁸O-temperature slopes through linear regression analysis between regional averages of simulated annual mean temperature and precipitation-weighted δ¹⁸O (Table 1). These regional δ¹⁸O-temperature regressions were applied to the regional ice core composites ($\delta^{18}O_{region}$ anomalies) to scale them from δ¹⁸O to temperature units (Figures 4 and 5), and produce temperature anomalies ($T_{region}$).

$$T_{region} = \alpha_{region}\delta^{18}O_{region} = \alpha_{region}\sum_{sites\ i} w_i\,\delta^{18}O_i$$

In this equation, $w_i$ represents the weights assigned to each site $i$, and $\delta^{18}O_i$ the site δ¹⁸O anomalies in 5 or 10 year averaged records. The limited length of the observational period (1979-2013 CE), does not allow us to precisely estimate the slope $\alpha$ on 10-year averages, and we preferred to use 1-year anomalies, where the slopes are significant (Table 1), and apply these slopes on the 10-year binned composites. It implies that the interannual δ¹⁸O-temperature relationship comes from mechanisms that are also applicable to decadal scale variability. It is impossible to further test this hypothesis without longer independent
temperature records. The use of the ECHAM-wiso isotope enabled climate model is the most up-to-date tool we have to quantify the δ¹⁸O-temperature on broad spatial-temporal scales, and is our best tool to infer the δ¹⁸O-temperature relationship in the absence of data. Its main limitation is the model resolution: it is missing some coastal topographical features, notably James Ross Island, Roosevelt Island, and Law Dome, and cannot faithfully represent regions where these sites are important. We refer to this scaling as the "ECHAM" method in the figure captions.

**Table 1:** Linear regression analysis (slope, correlation coefficient r, and p-value) of the simulated δ¹⁸O-temperature relationships extracted from the ECHAM5-wiso model for each climatic region, as well as the broad East Antarctic, West Antarctic and whole Antarctic regions.

| Geographic region | slope | r | p-value |
|---|---|---|---|
| 1. East Antarctic Plateau | 0.66 | 0.62 | <0.001 |
| 2. Wilkes Land Coast | 0.23 | 0.44 | 0.0084 |
| 3. Weddell Sea Coast | 0.34 | 0.34 | 0.0449 |
| 4. Antarctic Peninsula | 0.13 | 0.31 | 0.0658 |
| 5. West Antarctic Ice Sheet | 0.57 | 0.59 | <0.001 |
| 6. Victoria Land Coast | 0.59 | 0.49 | 0.00271 |
| 7. Dronning Maud Land Coast | 0.36 | 0.39 | 0.0271 |
| East Antarctica | 0.58 | 0.58 | <0.001 |
| West Antarctica | 0.55 | 0.56 | <0.001 |
| All Antarctica | 0.60 | 0.62 | <0.001 |

All regional δ¹⁸O temperature relationships produced by the ECHAM-wiso output are statistically significant (at 95% confidence) with the exception of the Antarctic Peninsula. Weak correlations are also found for the Weddell Sea coast (r = 0.34). Stronger correlation coefficients are obtained inland, for the larger-scale East and West Antarctic sectors, and maximum
values (r=0.62) are identified at the whole Antarctic scale.

Similarly, the simulated regional δ¹⁸O-temperature slopes are highest for the inland plateau (0.66 ‰ °C⁻¹) and lowest for the Peninsula (0.13 ‰ °C⁻¹). These low slopes for the Antarctic Peninsula do not agree with temporal δ¹⁸O-temperature relationships that have been reported for the highly-resolved James Ross Island (0.86 ‰ °C⁻¹; Abram et al., 2011), Gomez (0.5





‰ °C$^{-1}$; Thomas et al., 2009) ice cores and for precipitation samples collected at the O'Higgins Station (0.41‰ °C$^{-1}$, Fernandoy et al., 2012). ECHAM finds that the only site on the peninsula with a significant correlation between δ$^{18}$O and temperature is Bruce Plateau (66°S, 64°W) (slope=0.63±0.58‰ °C$^{-1}$, r=0.13, p=0.03), and the overall low δ$^{18}$O-temperature slope is largely attributable to model resolution. We expect that the ECHAM scaling will produce a temperature reconstruction with a low-amplitude bias in the Antarctic Peninsula.

High slopes similar to the inland East Antarctic plateau are identified in Victoria Land and West Antarctic Ice Sheet regions (0.59 and 0.57 ‰ °C$^{-1}$, respectively), together with intermediate values in coastal regions (i.e. coastal Indian, Weddell Sea and Dronning Maud Land regions) with a 0.31‰ °C$^{-1}$ mean slope. At the whole Antarctic ice sheet scale, the overall temporal slope is dominated by inland regions and simulated at 0.60 ‰ °C$^{-1}$. This analysis is more thoroughly examined in a study comparing an isotopic dataset from surface snow, snowfalls and ice cores (Goursaud et al., in prep).

### 3.4.2. Scaling based on NB2014 variance

In addition, we used an independent method of scaling the normalized δ$^{18}$O anomalies to the standard deviation ($\sigma(T)$) of the regional temperature from NB2014, over the 1960-1990 CE interval for the 5-year binned averages, and the period 1960-2010 CE for the 10-year binned averages. This scaling is similar to the one used for the CPS method (see next section).

$$T_{region} = \sigma(T)_{region}\delta^{18}O_{region}(normalized)$$

This scaling method implies that the δ$^{18}$O-temperature relationship can be inferred from the ratio of temperature to δ$^{18}$O standard deviation, which would be true if the relationship between the two was purely linear. If some of the δ$^{18}$O variance is due to something else than temperature, this scaling will under-estimate temperature variations. This method also assumes that the last 30 to 50 years provide a good estimate of the 5 or 10-year temperature variance. In the absence of longer temperature reconstructions, this is the best estimate of $\sigma(T)_{region}$ that we can provide.

We refer to this scaling as the "NB2014" method in the figure captions.

### 3.4.3. Scaling based on borehole temperature for the WAIS region.

In the West Antarctic Ice Sheet region, the approaches described above give different results (Figs 4 and 5), with the first method (temperature scaling from the δ$^{18}$O-T relationship in ECHAM-wiso) giving a smaller amplitude. At WAIS–Divide, there is an independent temperature record, which can be used to scale the long-term temperature evolution. We used the borehole temperature reconstruction at WAIS Divide (Orsi et al, 2012) to adjust the amplitude of temperature variations, matching the cooling trend over the period 1000-1600 CE (-1.1°C.1000y$^{-1}$). This scaling is actually in line with the scaling inferred from the NB2014 variance over 1960-2010 CE. The temperature calibration presented here is the best estimate we can provide with current knowledge, but we expect it to be revised in the future, with more precise δ$^{18}$O modeling, and more independent quantitative temperature reconstructions.

We refer to this scaling as the "borehole" method in the figure captions.

### 3.4.4 Replication of the 2013 Antarctica2k reconstruction method

To facilitate comparison with the preceding Antarctica2k temperature reconstruction published by the PAGES 2k Consortium (2013), we apply their reconstruction method to the updated ice-core isotope data collection. The method is a simple Composite-Plus-Scale (CPS) approach (Jones et al., 2009), updated from Schneider et al. (2006) and implemented similarly to Neukom et al. (2014). We apply this method to all sub-regions defined above, the broad East and West Antarctic divisions and to the Antarctic-wide database, replicating the Antarctic reconstructions presented in the PAGES 2k Consortium (2013) study.

First, the annual-average records are allocated to the climatic regions as defined above. The following steps are then repeated for each region separately. Second, only the records with no missing values in the 1961-1991 CE calibration period are selected.



These records are then scaled to mean zero and unit standard deviation over their common interval of data availability. Next, the normalised records are correlated against the NB2014 regional mean temperatures over 1961-1991 CE. Between 0% and 33% of the ice core records within each region have negative correlations (physically implausible) with the target and are removed from the proxy matrix. A composite of the remaining records is then calculated by creating a weighted average, where

the weighting of each ice core record is based on its temperature correlation from the previous step. The composite is then scaled to the mean and standard deviation of the NB2014 regional temperatures over the 1961-1991 CE period. The compositing and scaling steps are done in a nested approach, i.e. repeated for all periods with different proxy data availability. In the reconstruction of the PAGES 2k Consortium (2013), three records were infilled with neighbouring sites to have no missing data in the calibration window: WDC06A was infilled with data from WDC05A, Siple Station and Plateau Remote

were infilled by least median of squares multiple linear regression using nearby records (PAGES 2k Consortium, 2013). To allow comparison, we also used the infilled data for these records. Thus, the only difference to the reconstruction of PAGES 2k Consortium (2013) is that we use an extended proxy database with more records and an updated temperature target (NB2014 instead of Steig et al. 2009).

While this CPS approach allows a quantitative calibration to the NB2014 temperature data, it has some limitations compared

to the above methods. First, in this implementation it allows only the inclusion of data covering the calibration period, thereby removing more than half of the available records (62 out of 112). Second, the calibration period is extremely short and therefore individual years (for example with outliers) can significantly bias the reconstruction and reasonable validation of the reconstruction is hardly possible. The main difference from the other compositing methods described above is the weight of each record and the interval, over which the data are standardized.

**4 Results and Discussion**

We use the varying reconstruction methods to identify robust trends in the Antarctic ice core database. We present results based on isotopic trends, as well as temperature reconstructions, and examine these for the seven climatic regions and for the larger-scale Antarctic regions (Section 4.1); compare our results to the previous Antarctic temperature reconstruction (Section 4.2), and investigate the link between temperature and volcanic activity (Section 4.3).

**4.1 Regional-scale $\delta^{18}$O and temperature reconstructions**

Five year-binned $\delta^{18}$O composite records since 1800 CE years are reported for each of the seven climatic regions in Supplementary Figure 2 (unweighted isotope anomalies) and Figure 2 (weighted and unweighted data). The unweighted composites are shown with respect to the distributions of data within each bin, and expressed relative to the 1960-1990 CE interval. Figure 2 also shows the reconstructions that are obtained by weighting the records within each region based on the

NB2014 temperature field and by the ECHAM5-wiso $\delta^{18}$O field. Figure 3 (and Supplementary Figure 3) shows equivalent data, but for 10y averages since 0 CE relative to the 1900-1990 CE interval.

The highest density of ice core records is present in the last century, but these are not evenly distributed over Antarctica (Supplementary Figure 1) with most of the records in the plateau and coastal areas of Dronning Maud Land and across the West Antarctic Ice Sheet. On the other hand, only 1 and 3 records are present in the Weddell Sea and Wilkes Land coastal

areas, respectively.

In order to separate the uncertainties dues to the stacking procedure from uncertainties in the temperature scaling, we first discuss the main features of the un-weighted regional $\delta^{18}$O anomalies (Section 4.1.1), and then proceed to discuss the weighted regional $\delta^{18}$O anomalies, and finally the temperature reconstructions. The weighted and unweighted composites produce similar results for the seven climatic regions (see figures in Supplementary Information from S4 to S10), suggesting that our

reconstructions are robust, and don't depend on the exact methodology used (Figures 2 and 3). There are two exceptions to





this: the Antarctic Plateau (Fig. S4), where the many sites in Dronning Maud Land perhaps still bias the simple average towards this area even with the gridded data-reduction process, and the West Antarctic ice Sheet (Fig. S8) where the two long records from WAIS-Divide and Roosevelt Island (RICE) have diverging trends for much of the last 2000 years. The weighted reconstruction gives a higher weight to WAIS-Divide, which maintains a long-term negative isotopic trend over the last 2ka
in this region (see section 4.1.2). We further checked that WAIS-Divide is indeed more representative of temperatures averaged across the West Antarctic Ice Sheet region than RICE looking at temperature correlation maps which use the NB2014 temperature field (Fig. S8). The temperature reconstructions obtained with the different methods are further shown in Fig. S15. We assess trends for the seven climatic regions (and in the larger-scale Antarctic groupings) for the reconstructions prior to 1900 CE (up to 1,900 years length) (Section 4.1.2), and since 1900 CE (up to 110 years length) (Section 4.1.3), and estimate
the significance of the most recent 100-year trend relative to natural variability (Section 4.1.4).

### 4.1.1 Trend significance in unweighted composites

We first use a Monte Carlo approach to assess the significance of trends in the unweighted composites. This test is designed to test the significance of trends in relation to the distribution of data within each bin of the isotopic composites. For each bin
where 2 or more ice cores contribute data we scale random Gaussian data about the median value and $\pm2\sigma$ distribution of isotopic data within that bin. We then sample from this scaled Gaussian data to produce 10,000 simulations of each regional composite. We then assess the proportion of ensemble members which produce trends of the same sign as the mean composite, and the proportion of ensemble members where the trend is of the same sign as the mean composite and also significant at greater than 95% confidence. These trend analyses are based on 10-year binned isotopic anomalies for trends prior to 1900
CE, and 5-year binned data for trends over the last 100 years of the composites (although equivalent results are found if 10-year binned data are used to assess trends in the last 100 years).

Results of unweighted composite trend analysis are summarised in Table 2. This analysis shows that for the unweighted composites the long-term cooling trend from 0-1900 CE is only significant for the East Antarctic plateau. Visual examination of the unweighted composites (Figure 3) suggest that many of the other climate regions appear to also have a negative
unweighted isotopic trend over part of the last 2000 years (e.g. Wilkes Land and West Antarctic Ice Sheet regions), but these trends are not significant in the unweighted composites when calculated across the full interval from 0-1900 CE. The Victoria Land coast trend prior to 1900 CE is not significant in the mean or median unweighted composites, but over the interval where two or more ice cores contribute to the composites the Monte Carlo assessments indicate that negative isotopic trends are produced in all 10,000 ensemble members and are significant ($p<0.05$) more often than can be explained by chance alone.
The significant trend that is produced in the unweighted composite for the East Antarctic Plateau climate region is also evident in in the broader East Antarctic compilation and for the Antarctic continent-scale composite. The continent-scale cooling trend produced in unweighted composites using the expanded Antarctica 2k database is in agreement with the PAGES 2k Consortium (2013) results where a long-term cooling trend over the Antarctic continent was identified. It is also consistent with robust pre-industrial cooling trends that have been identified in other continental reconstructions (PAGES 2k Consortium)
and in the global oceans (McGregor et al 2015).

Isotopic trends in the last 100 years of the unweighted composites show significant positive trends across a number of regions. In particular, significant isotopic trends, indicative of climate warming, are evident in the unweighted composites for the Antarctic Peninsula, and the Wilkes Land and Dronning Maud Land coasts. The West Antarctic Ice Sheet region does not display a significant ($p<0.05$) positive trend in the mean or median of the unweighted composites, but the Monte Carlo analysis
across the distribution of isotopic data within each 10-year bin suggests that positive trends are produced in 99.5% of the 10,000 simulations and are significant ($p<0.05$) more often than can be explained by chance alone (20.5% of simulations). Similarly, the Victoria Land-Ross Sea region shows a negative but insignificant trend in the mean and median composites, but in the Monte Carlo simulations this negative (cooling) trend is produced in 99.9% of ensemble members and is significant in



13.6% of ensemble members. The apparent inverse isotopic trends during the last century between the Victoria Land-Ross Sea region and the West Antarctic Ice Sheet and Antarctic Peninsula regions may be indicative of dynamical processes in the Amundsen Sea sector, which on interannual time scales are known to cause opposing climate anomalies between these regions. The PAGES 2k Consortium (2013) study concluded that Antarctica was the only continent-scale region to not see of the long-

term cooling trend of the past 2000 years reverse to recent significant warming. However, at the regional scale examined in this study recent significant warming is suggested by many of the unweighted isotopic composites, particularly for coastal regions of Antarctica and the West Antarctic ice sheet. It should be stressed that these results are based only upon the simple unweighted compositing of the regional isotopic data, and the significance of trends is further assessed in the following section after weighting of the individual ice core records based on how representative they are of isotopic and temperature variability

within each climatic region.

Table 2: Summary of trend statistics for isotopic anomalies using unweighted composites. Trends are expressed as isotopic anomalies, in per mil $\delta^{18}O$ per decade units. Trends and their significance (p; based on the Student's t-statistic) are calculated using a linear additive model, and reported for the mean and median composites. Monte Carlo testing is applied to 10,000

ensemble members based on the unweighted composite distributions, which are assessed to determine the percentage of trends with the same sign as the mean trend, and the percentage with the same sign and a significance of p < 0.05. Bold values indicate mean and median trends with a significance of p < 0.05, and Monte Carlo simulations where significant, same-signed trends exceed 5% of the ensemble.

| Region | Pre-1900 CE trends | | | Last 100y trends | | |
|---|---|---|---|---|---|---|
| | Mean trend ‰ 10y⁻¹ (p) | Median trend ‰ 10y⁻¹ (p) | Monte Carlo % (% p<0.05) | Mean trend ‰ 10y⁻¹ (p) | Median trend ‰ 10y⁻¹ (p) | Monte Carlo % (% p<0.05) |
| East Antarctic Plateau | **-0.003 (0.000)** | **-0.002 (0.000)** | **100 (99.2)** | 0.011 (0.675) | 0.040 (0.096) | **96.2 (14.1)** |
| Wilkes Land coast | 0.017 (0.444) | 0.017 (0.444) | 80.6 (2.3) | **0.098 (0.001)** | **0.098 (0.001)** | **100 (90.5)** |
| Weddell Sea coast | -0.002 (0.376) | -0.002 (0.376) | - | -0.060 (0.318) | -0.060 (0.318) | - |
| Antarctic Peninsula | 0.006 (0.188) | 0.004 (0.361) | 84.3 (2.9) | **0.112 (0.000)** | **0.111 (0.000)** | **100 (98.2)** |
| West Antarctic ice sheet | 0.000 (0.435) | 0.000 (0.428) | 85.4 (1.4) | 0.033 (0.226) | 0.042 (0.091) | **97.0 (14.4)** |
| Victoria Land-Ross Sea | -0.006 (0.118) | -0.006 (0.101) | **100 (12.76)** | -0.054 (0.220) | -0.074 (0.102) | **99.9 (13.6)** |
| Dronning Maud Land | -0.032 (0.366) | -0.027 (0.448) | 95.3 (0.1) | **0.147 (0.000)** | **0.158 (0.001)** | **100 (99.8)** |
| East Antarctica | **-0.003 (0.000)** | **-0.002 (0.000)** | **100 (100)** | **0.037 (0.035)** | **0.064 (0.003)** | **99.8 (55.1)** |
| West Antarctica | 0.000 (0.437) | -0.001 (0.118) | **95.57 (11.2)** | **0.054 (0.021)** | **0.082 (0.001)** | **100 (79.8)** |
| All Antarctica | **-0.002 (0.000)** | **-0.002 (0.000)** | **100 (99.5)** | **0.044 (0.005)** | **0.073 (0.000)** | **99.9 (76.2)** |

### 4.1.2 Long term trends in weighted reconstructions

To extend this trend analysis further we next assess the pre-1900 CE (Figure 5) trends in the temperature reconstructions

produced by scaling the ice core data based on our different approaches (see section 3.3.3). We estimate the uncertainty of the slope based on the ±2σ uncertainty in the regression parameters. The robustness of the slope estimation to individual data points was further checked by taking out 10% of data points randomly and calculating the slope again, but the uncertainty estimates this yields is much smaller than the uncertainty based on regression parameters. The slopes obtained by each of the temperature scaling methods are presented in Table 3. The uncertainty in the amplitude of the 0-1900 CE trend is dominated

by the uncertainty in the temperature scaling of the composite. To make the discussion clear, we first focus on the trend in terms of normalised anomalies, with respect to the 1900-1990 CE periods, which circumvents the temperature scaling issues. The period 0-1900 CE exhibits significant negative trends in most regions, from -0.4 to -1.32σ 1000y⁻¹. The trend is largest in West Antarctic Ice Sheet (-1.32±0.23σ 1000y⁻¹), Victoria Land (-0.89±0.39σ 1000y⁻¹), and the Antarctic Plateau (-





$0.76\pm0.28\sigma$ $1000y^{-1}$) regions. It is smaller but still significant for the Wilkes Land coast ($-0.59\pm0.48\sigma$ $1000y^{-1}$, p=0.007) and Antarctic Peninsula ($-0.50\pm0.23\sigma$ $1000y^{-1}$). It is insignificant on the Weddell Sea coast (p=0.3), and the dataset is not long enough to estimate millennial-scale trends on the DML coast. These observations indicate a broad scale cooling trend over the continent, that is comparable in amplitude to the variance over the past 100 years. As previously mentioned, the 0-1900 CE

negative trend is largest in the normalized datasets for the West Antarctic Ice Sheet region, but this feature masks sub-regional scale differences, with the RICE record indicating increasing rather than decreasing anomalies. This result is puzzling because both WAIS Divide (to the west) and Victoria Land cores (e.g. VLG, to the east of RICE) show a clear cooling trend prior to 1900 CE, and together this points to a more complex picture of the evolution of the atmospheric circulation around the Ross Sea than can be captured with the current network of cores. The RICE $\delta^{18}O$ record, situated on Roosevelt Island on the Ross

Ice Shelf, is more influenced by air masses from the eastern Ross Sea sector than the rest of the West Antarctic Ice Sheet region which is influences predominantly by air masses from the Amundsen Sea (Neff et al., 2015; Emanuelsson et al. in review). Moreover, borehole temperature and $\delta^{15}N$ thermal fractionation records at RICE highlight some notable differences to the isotope temperature reconstruction, which suggest that sea ice extent exerts an important control, perhaps masking aspects of the longer term temperature trends in the region (Bertler et al., in preparation).

Over the period from 0-1900 CE the ECHAM temperature scaling suggests that the mean cooling trends are $-0.2^{\circ}C$ $1000y^{-1}$ for the east Antarctic Plateau and Antarctic Peninsula, $-0.3^{\circ}C$ $1000y^{-1}$ for the West Antarctic ice sheet and $-0.2^{\circ}C$ $1000y^{-1}$ for Victoria Land. Coastal East Antarctic regions do not show significant trends (Weddell Sea, Dronning Maud Land coast and Wilkes Land coast). The NB2014 temperature scaling produces 50% larger trends overall (Table 3), and triples the values for the West Antarctic Ice Sheet ($-0.9^{\circ}C$ $1000y^{-1}$). The trend in the West Antarctic Ice Sheet region can be assessed independently

by comparing the WAIS composite to the borehole temperature record of Orsi et al. (2012) at WAIS-Divide for the period 1000-1600 CE. The borehole-derived trend at WAIS divide is $-1.1^{\circ}C$ $1000y^{-1}$, close to the $-1.0^{\circ}C$ $1000y^{-1}$ value found with the NB2014 method over 1000-1600 CE, but much larger than $-0.38^{\circ}C.1000y^{-1}$ found with the ECHAM scaling. The CPS method finds a slope of $-0.72^{\circ}C$ $1000y^{-1}$ for the period 1000-1600 CE, slightly lower than what is inferred from the borehole temperature reconstruction.

Over all of Antarctica, we find a cooling trend of $-0.2$ to $-0.4^{\circ}C$ $1000y^{-1}$ between 0-1900 CE across the various scaling approaches (Table 3). This trend is comparable in magnitude of the $-0.22\pm0.06^{\circ}C$ $1000y^{-1}$ found for the Arctic region (Kaufman et al, 2009). This trend was attributed to a decrease in JJA solar insolation due to precession of the equinoxes.

**Table 3:** Trend analysis of the period 0-1900 CE (or shorter depending on maximum record length), for the various temperature scaling approaches detailed in Section 3.4.

| Region | interval | | ECHAM | | NB2014 | | Borehole | |
|---|---|---|---|---|---|---|---|---|
| | start year | end year | Slope (°C 1000y⁻¹) | p-value | Slope (°C 1000y⁻¹) | p-value | Slope (°C 1000y⁻¹) | p-value |
| 1. Plateau | 0 | 1900 | -0.24±0.09 | 0.000 | -0.32±0.12 | 0.000 | NaN | NaN |
| 2. Wilkes Coast | 170 | 1900 | -0.11±0.08 | 0.007 | -0.16±0.13 | 0.016 | NaN | NaN |
| 3. Weddel coast | 1000 | 1900 | -0.08±0.19 | 0.376 | -0.14±0.31 | 0.376 | NaN | NaN |
| 4. Peninsula | 0 | 1900 | -0.16±0.07 | 0.000 | -0.20±0.09 | 0.000 | NaN | NaN |
| 5. WAIS | 0 | 1900 | -0.28±0.06 | 0.000 | -0.92±0.16 | 0.000 | -0.85±0.18 | 0.000 |
| 6. Victoria Ross | 0 | 1900 | -0.17±0.09 | 0.000 | -0.29±0.12 | 0.000 | NaN | NaN |
| 7. DML coast | 1530 | 1900 | 2.30±1.27 | 0.001 | 0.97±0.97 | 0.003 | NaN | NaN |
| West Antarctica | 0 | 1900 | -0.15±0.04 | 0.000 | -0.44±0.09 | 0.000 | -0.75±0.15 | 0.000 |
| East Antarctica | 0 | 1900 | -0.18±0.06 | 0.000 | -0.32±0.10 | 0.000 | NaN | NaN |
| All Antarctica | 0 | 1900 | -0.20±0.04 | 0.000 | -0.40±0.08 | 0.000 | -0.15±0.03 | 0.000 |



| | normalized | | CPS | | | |
|---|---|---|---|---|---|---|
| Region | slope (σ1000 y⁻¹) | p-value | slope | p-value | Start year | End year |
| 1. Plateau | -0.76±0.28 | 0.000 | -0.15±0.06 | 0.000 | 10 | 1900 |
| 2. Wilkes Coast | -0.59±0.48 | 0.016 | -0.10±0.07 | 0.008 | 180 | 1900 |
| 3. Weddel coast | -0.41±0.91 | 0.376 | -0.09±0.27 | 0.494 | 1000 | 1900 |
| 4. Peninsula | -0.50±0.23 | 0.000 | -0.09±0.09 | 0.048 | 0 | 1900 |
| 5. WAIS | -1.32±0.23 | 0.000 | -0.59±0.08 | 0.000 | 0 | 1900 |
| 6. Victoria Ross | -0.89±0.39 | 0.000 | -0.54±0.58 | 0.066 | 1140 | 1900 |
| 7. DML coast | 4.98±3.2 | 0.003 | NaN | NaN | NaN | NaN |
| West Antarctica | -0.76±0.16 | 0.000 | -0.55±0.55 | 0.000 | 0 | 1900 |
| East Antarctica | -0.59±0.19 | 0.000 | -0.18±0.06 | 0.000 | 10 | 1900 |
| All Antarctica | -0.76±0.15 | 0.000 | -0.38±0.05 | 0.000 | 0 | 1900 |

### 4.1.3 Trends of the last 100 years in weighted reconstructions

Studies based on individual ice cores have identified significant positive trends in the last century (Mulvaney et al., 2012; Steig

5 et al., 2013). Similar to the findings for unweighted composites, significant warming trends in the last 100 years (Table 4) of the weighted anomalies are evident for the Antarctic Peninsula (+2.65σ 100y⁻¹ with respect to the 1960-1990 CE normalisation period), Dronning Maud Land coast (+2.51σ 100y⁻¹) and the West Antarctic Ice Sheet (+1.17σ 100y⁻¹) regions. The trends in other regions are not significant (Table 4). In temperature units, the NB2014 method gives a scaling about twice as ECHAM, while CPS is in between (Table 4). Since 1900CE, the reconstructions indicate that the Antarctic Peninsula has been warming

10 by 0.9-1.99°C 100y⁻¹, West Antarctic Ice Sheet by 0.66-2.0°C 100y⁻¹, and the Dronning Maud Land coast by 0.51 to 0.98°C 100y⁻¹. The borehole temperature adjustment needed to match the long term trend leads to an over-estimation of the 100-year trend in the WAIS region. Indeed, the same borehole temperature record finds a warming trend of 0.70°C 100y⁻¹ for the past 100 years, and 1.22°C 50y⁻¹ for the past 50 years. This example shows that a simple linear scaling cannot be valid for all timescales, and that another approach will be needed to improve quantitative temperature reconstructions for Antarctica.

15 Despite these uncertainties on absolute scaling, our analyses underline that the sustained warming of the Antarctic Peninsula over the last century stands out as being a robust feature across all methods. Moreover, while the West Antarctic Ice Sheet and the Peninsula regions have now seen reversal of the long-term cooling trend of the past 2000 years, this is not the case for the rest of the continent, where temperatures changes over the last century have not been significant (Figure 4, Table 4).

**Table 4:** Regression of the last 100-year slope based on the 5-year averages

| | normalized | | ECHAM | | NB2014 | Borehole | CPS | |
|---|---|---|---|---|---|---|---|---|
| Region | Slope (σ 100y⁻¹) | p-value | Slope (°C 100y⁻¹) | p-value | Slope (°C 100y⁻¹) | Slope (°C 100y⁻¹) | slope | p-value |
| 1. Plateau | -0.85 | 0.133 | -0.31 | 0.174 | -1.28 | NaN | 0.66 | 0.024 |
| 2. Wilkes Coast | 1.08 | 0.115 | **0.75** | **0.034** | 0.47 | NaN | 0.15 | 0.586 |
| 3. Weddel coast | -1.10 | 0.351 | -0.27 | 0.351 | -0.50 | NaN | -0.27 | 0.608 |
| 4. Peninsula | **2.65** | **0.000** | **0.90** | **0.000** | 1.99 | NaN | **1.42** | **0.001** |
| 5. WAIS | **1.17** | **0.014** | **0.66** | **0.020** | 0.97 | 2.03 | **0.44** | **0.239** |
| 6. Victoria Ross | -0.79 | 0.195 | -0.27 | 0.247 | -0.64 | NaN | 0.05 | 0.845 |
| 7. DML coast | **2.51** | **0.012** | **0.51** | **0.006** | 0.98 | NaN | **0.74** | **0.050** |
| West Antarctica | **1.50** | **0.001** | **0.78** | **0.007** | **1.12** | **1.85** | **0.63** | **0.091** |
| East Antarctica | 0.04 | 0.924 | 0.05 | 0.743 | 0.06 | NaN | 0.32 | 0.082 |
| All Antarctica | **0.72** | **0.040** | 0.45 | 0.060 | 0.99 | 0.21 | **0.40** | **0.049** |





### 4.1.4 Significance of most recent 100y trends relative to natural variability

Finally, we assess how significant the trends in the most recent 100-years of the regional temperature reconstructions are relative to the range of all other 100-year trends in the reconstructions since 0 CE (Figure 6). The most recent 100y trend and

its (±1σ) uncertainty range are compared to the distribution of all other 100y trends for that region. In this analysis the significant recent positive trends for the Dronning Maud Land coast and West Antarctic ice sheet do not emerge as unusual in the context of similar length trends of the last two millennia, consistent with the findings of Steig et al. (2013) based on the WAIS Divide ice core. However, the most recent 100-year warming of the Antarctic Peninsula is unusual compared to the range of natural century-scale warming trends of the last two millennia (Mulvaney et al., 2012). Thus across all of our trend

tests, the warming of the Antarctic Peninsula over the last century stands out as being robust to the binning of multiple isotopic anomaly records for the region, significant after weighting and scaling of the regional isotopic records to represent regional temperature, and unusual compared with the range of natural century-scale temperature variability of this region over the last two thousand years. The companion Antarctica 2k paper (Thomas et al., in review) that examines snow accumulation across Antarctica over the last 1000 years also concludes that the most robust recent changes in snowfall are evident for the Antarctic

Peninsula.

Prior to this recent warming, which has been most significant for the Antarctic Peninsula region of Antarctica, long-term cooling is evident for many regions of Antarctica. In particular, it is the long-term cooling of the East Antarctic Plateau which emerges as significant in the weighted reconstructions and also robust across the distribution of contributing records in the unweighted composites. This is also in agreement with what was recently found by Hakim et al. (2016) using a data

assimilation approach to identify a very strong SH cooling trend through the last 2000 years, up to about 1850 CE.

### 4.2 Continent-scale temperature reconstructions

Figure 7 shows the results obtained by applying the Composite-Plus-Scale (CPS) method published in 2013 (PAGES2k Consortium, 2013) to the expanded Antarctica2k database to reconstruct temperature across the whole Antarctic continent. A good agreement is observed between the new reconstruction and the previously published one (Figure 7), both at annual and

decadal scale (r=0.87 and 0.89, respectively). There is a slightly stronger negative temperature trend in the new Antarctic continent-scale reconstruction, which is related mainly to a slightly cooler temperature estimate for the 1600-1900 CE interval using the expanded database. Figures in supplementary material, similarly repeat this CPS comparison for the West Antarctica and East Antarctica regions and find similar agreement between the 2013 results and those using the expanded ice core database (Figs S11, S12, and S13 respectively).

Figure 8 shows the continental scale (All Antarctica), East Antarctica and the West Antarctic temperature reconstructions that have been obtained from the different temperature scaling approaches in comparison with the CPS method. The period between 1200 and 1900 CE appears as the coldest interval in the East Antarctic reconstruction, and this is also reflected in the reconstruction at the whole continental scale. The warmest interval is identified between 300 and 1000 CE, especially for West Antarctica. The comparison between the CPS method and our new reconstruction methods shows a better agreement in East

Antarctica than in West Antarctica when considering the ECHAM temperature scaling (Figure 8). However, a better agreement between the CPS method and the temperature reconstruction obtained with the borehole scaling method is observed for West Antarctica.

Over the period from 165-1900 CE (the reconstruction interval of the 2013 PAGES 2k reconstruction for Antarctica), a cooling trend is common to all three broad-scale reconstruction regions and across all reconstruction methods (see Table S2). Cooling

trends range between -0.13 and -0.26°C 1000y⁻¹ for the ECHAM method, and between -0.18 and -0.74°C 1000y⁻¹ for the borehole method. These values are in the range of those previously reported by the 2013 reconstruction (PAGES2k Consortium, 2013) where cooling trends of -0.18 and -0.46°C 1000y⁻¹ were calculated, for the same interval, for East and West





Antarctica, respectively. By comparison, cooling trends of -0.25 and -0.56°C 1000y$^{-1}$ are calculated from the 10y binned data obtained with the same CPS methods but applied to this new database.

East Antarctica is one of the last places on the globe where the long term cooling trend of the last millennium has not been inverted (PAGES2k Consortium, 2013; Abram et al., 2016). This feature of delayed onset of industrial warming is clear in our

reconstruction, but is not captured by climate models (PAGES 2k–PMIP3 group, 2015). Climate models are known to overestimate inter-hemispheric synchronicity (Neukom et al., 2014). The model-data mismatch may come from an over-estimation of the response to external forcing in climate models, an under-estimation of the role of internal ocean-atmosphere dynamics in models, or incorrect inference from proxy data. This quantitative reconstruction will provide an important constraint on the forced vs unforced nature of trends observed for the last century.

**4.3 Response to volcanic forcing**

The radiative forcing caused by major volcanic eruptions is known to drive hemispheric-scale cooling at subdecadal to decadal timescale (e.g. Sigl et al., 2015; Stoffel et al., 2015). Previous model-based assessments have suggested that the long-term cooling trend over Antarctica during the 850-1850 CE interval may be attributed to volcanic forcing of the climate (Goosse et al., 2012). Volcanic forcing may also affect atmospheric and oceanic dynamics, including modes of variability, causing

substantial seasonal and regional variations in the climate footprint of volcanic forcing (e.g. Otterå et al., 2010; Schleussner et al., 2015; Ortega et al, 2015; Swingedouw et al, 2015). For example, the long-term cooling trend of the global oceans during the last two millennia has been attributed to the effects of short-term cooling from episodic eruptions being cycled into subsurface layers of the oceans (McGregor et al., 2015).

We use the composite reconstructions of Antarctic temperature anomalies to explore whether a signatures/imprints of past

major eruptions can be identified. Since different normalizations applied during the reconstruction procedure are not expected to affect a qualitative interpretation of the results, we present the analysis based on the reconstruction that fits 5-year annual $\delta^{18}$O to regional temperature anomaly from ECHAM-wiso forced by ERA-Interim.

A robust identification of the climate impacts of volcanism from Antarctic ice cores faces a number of challenges. Low accumulation and wind driven post-depositional processes are limitations to seasonal layer counting of stable water isotopes

and/or chemical profiles, and to climatic signals retrieved from single cores. In addition to the poor chronological constraints, processes other than local temperature may affect water stable isotopes in Antarctic precipitation, therefore making the detection of a temperature response to volcanic forcing more challenging. Finally, stacking the individual core records acts as a low-pass filter, suppressing the variability in the shorter, interannual timescales, thereby eliminating potentially short-lived cooling signals triggered by volcanic forcing.

Despite these limitations, it should be possible to record a volcanic cooling signal from well-dated cores from high accumulation areas and/or around the time periods with clear markers of volcanic origin, widely used as precise age control points for constructing ice core chronologies. For example, the chemical fingerprints of two consecutive tropical eruptions of 1808/09 (VEI 6-7) and Tambora 1815 (VEI 7) are detected in many Antarctic ice cores as a pronounced double peak in a number of measured ice parameters such as electrical conductivity or non-sea salt-SO$_4^{2-}$ profiles (Sigl et al., 2015).

Due to the aforementioned difficulties in the identification of a potential Antarctic temperature response to volcanic forcing, we restrict our analysis to the eruptions associated with the estimated strongest volcanic forcing, which are more likely to leave a clear footprint in ice core records. The highest resolution reconstruction (5 year averages) is available for a period which experienced only three eruptions with a Volcanic Explosivity Index (VEI) of 6-7, namely the tropical eruptions of 1808/09, 1815 Tambora and 1883 Krakatoa. The signatures of the first two eruptions are commonly used as reference horizons when

constructing the core chronologies. Due to a low number of replicates, we cannot use a Superposed Epoch Analysis, commonly applied for detecting the signature of the climate response to volcanic forcing (e.g. Fischer et al., 2007; Stoffel et al, 2015).



Figure 9 simply displays the regional surface atmospheric temperature anomalies for the two periods centred at 1815 CE and 1883 CE.

Following the Tambora eruption, cooling is recorded in 5 out of our 7 regional reconstructions, namely in the Dronning Maud Land coast, Wilkes Land coast, Victoria Land coast, East Antarctic Plateau and possibly West Antarctic Ice Sheet regions.

Following the 1808/09 eruption, cooling appears in 3 out of our 7 regional reconstructions (West Antarctic Ice Sheet, Wilkes Land coast regions and to a lesser extent the Antarctic Peninsula). Following the Krakatoa eruption, regional cooling emerges for 4 out of 7 regions (East Antarctic Plateau, Antarctic Peninsula, Wilkes Land coast and West Antarctic Ice Sheet regions). Potential response time lags should be interpreted with caution due to effects of the data binning procedure as well as age scale uncertainties.

Within the limitations associated with our reconstructions and with the small number of large eruptions in the studied time period, we can only report a lack of consistent pan-Antarctic cooling, and a variable regional response. Such features have also been identified for the largest volcanic eruptions in the Northern Hemisphere (e.g. Guillet et al., in press). Local negative anomalies following major eruptions may arise from non-temperature drivers of water stable isotope records and be an artefact of inconsistencies in chronologies; they may also reflect an actual regional cooling. Further studies will require to focus on ice

core records with a solid chronological control, and sufficient regional records to assess the common climatic signal from the deposition noise. Second order isotopic parameters such as deuterium excess which preserve a signal associated with moisture source characteristics may also provide insights on the response of the Southern Ocean surface state to volcanic forcing.

This analysis represents only a preliminary attempt to assess the possible climatic response of Antarctic climate to short-term volcanic eruptions, and we suggest that this is a good avenue for further detailed study using the expanded Antarctica 2k

isotope database assembled here.

**5 Conclusions and Implications**

We have presented a new enlarged database of high and low resolution ice core water stable isotope records from Antarctica, which has been compiled in the framework of the PAGES Antarctica2k working group. To further develop our scientific understanding based on continental scale reconstructions of Antarctic temperature, we define seven climatic regions based on

our knowledge of the regional Antarctic climate and further supported by spatial correlation of temperature using the NB2014 reanalysis data. In our reconstructions of both regional and continental scale climate, we use 5y and 10y averaged data in order to limit our dependence on annually precise dating which faces real problems, especially in interior Antarctica, due to the very low snow accumulation there. Of the initial 112 records, 79 and 67 records meet our minimum requirement of having at least 30y or 90y data coverage since 1800 or 0 CE, respectively. We produce simple unweighted and weighted composites of ice

core $\delta^{18}$O anomalies, as well as weighted temperature reconstructions which take into account the regression between each ice core site and the regional temperature. Different weighting methods and temperature scaling performed using the NB2014 temperature product as well as the ECHAM temperature and isotopic fields support the robustness of our reconstructions. Replication of a CPS approach used in a previous continental scale reconstruction but adapted to the new database also produces consistent results.

Our new continental scale reconstructions, based on the extended database, corroborates previously published findings for Antarctica from the PAGES2k Consortium (2013): (1) Temperatures over the Antarctic continent show an overall cooling trend during the period from 0 to 1900CE, which appears strongest in West Antarctica, and (2) no continent-scale warming of Antarctic temperature is evident in the last century.

The robust and significant long-term cooling trend of Antarctic-wide temperatures from 0-1900 CE is also reflected in the

broader East Antarctic and West Antarctic regions. In East Antarctica the coldest interval occurs from 1200 and 1900 CE. During the same time period five intervals of volcanic-solar downturns have been previously linked to cooling in temperature



reconstructions for other continental regions (PAGES2k Consortium, 2013). These intervals of negative radiative forcings are in agreement with our lower temperature estimates during this time (Fig.8). A test of volcanic impact performed on our 200y regional temperature reconstructions, although not conclusive due to the limit imposed by our 5y binned data set, has nevertheless showed possible negative temperature effects in some regions, in particular the East Antarctic Plateau and some

coastal regions. More research on this topic with a special focus on individual records with limited age uncertainties and high-resolution data sets would be needed for reaching a more conclusive answer and could be facilitated using the expanded Antarctic 2k database.

The absence of significant continent-scale warming of Antarctica over the last 100 years is in clear contrast with the significant industrial era warming trends that are evident in reconstructions for all other continents (except Africa) and the tropical oceans

(Abram et al., 2016). As noted in other studies (e.g. Abram et al., 2016; Jones et al., 2016) the absence of continent-scale warming over Antarctica is not in agreement with climate model simulations, which consistently produce a 20[th] Century warming trend over Antarctica in response to greenhouse gas forcing. The high interannual climate variability with respect to the magnitude of climate trends over Antarctica, and the small number of ice core records that have contributed to previous continent-scale reconstructions of Antarctic temperature, have been suggested as possible sources for the lack of detection of

continent-scale warming trends in palaeoclimate records from Antarctica. However, we replicate this finding of a continent-scale absence of 20[th] Century warming using our greatly expanded ice core database and with composites and reconstructions based on a range of methods. This suggests that the absence of continent scale warming over Antarctica during the last century is a robust result and alternate reasons for data-model divergences need to be investigated.

Despite the lack significant warming over the last 100 years at the continent scale, there are three regions which show positive

and significant isotopic and temperature trends:  the Antarctic Peninsula, the West Antarctic Ice Sheet and the coastal Dronning Maud Land. However, currently only for the Antarctic Peninsula is the last 100 y trend unusual in the context of natural century-scale trends over the last 2000 years, although if sustained and significant warming of Dronning Maud Land coast and the West Antarctic ice sheet continues during the coming decades (as would be expected based on climate model simulations), then these areas may soon also see the recent warming emerge as unusual in the context of natural variability. A companion

paper (Thomas et al., in review), which examines the snow accumulation rate variability over Antarctica during the last millennium, reaches the same conclusion showing that in the Antarctic Peninsula region the late 20[th] century snowfall increase is unusual in the context of the past 300 years. Temperature and precipitation increases over the Antarctic Peninsula in recent decades have previously been linked to the Southern Annular Mode (Abram et al., 2014) and the effects of the coupling between SAM and ENSO on the strength and position of the Amundsen Sea Low pressure and sea ice conditions offshore of

West Antarctic and the Antarctic Peninsula (Thompson and Solomon, 2002; Ding et al., 2011; Thomas and Abram, 2016). The variability of these driving forcings over the past 2000 years could also explain part of the opposite regional climate variability that appears to be present at decadal-centennial scales between our reconstructions for the Antarctic Peninsula and the Victoria Land-Ross Sea regions. These regional to local scale patterns of climate changes, and the processes that these illustrate, are another target for future research that could be explored using the new Antarctica2k database. Future work will also consider

the comparison of the results obtained in this study with those by Thomas et al. (in review, same issue) assessing the snow accumulation rate variability in the same regions, with the aim of exploring a long-standing question about the relationship between temperature and precipitation in Antarctica.

Our new regional and continental scale reconstructions of Antarctic temperatures highlight again the urgent need to increase the spatial coverage of this sparse data continent with new temperature records, particularly focusing in the coastal areas. These

regions are typically seen to be data-poor areas in our extended database. However, the coastal areas are also the places where high-resolution ice cores can be retrieved, where strong climatic differences can occur over relatively short distances, where there is some evidence for non-significant but emerging development of warming trends, where even small amounts of

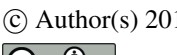



warming can rapidly move the local climate beyond the threshold and where surface melting of snow occurs potentially leading to widespread impacts of any future Antarctic warming.

**Supplementary Information**

Supplementary figures (from Fig. S1 to S15) and tables (from table S1 to S4) are available as Supplementary Information.

**Author contribution**

BS led Phase 2 of the Antarctica 2k project, and MAJC was data manager. The data analysis presented in this paper was carried out by NJA, AO, SG, RN and DD. BS, NJA and AO led the writing of the manuscript, with contributions by MAJC, SG, RN,

VM-D, HG, DD and EJS. All authors contributed to discussions on the analysis design and manuscript.

**Acknowledgements**

This manuscript is the Antarctic contribution to the international effort to characterise global climate variability during the last 2000 years, coordinated within the Past Global Changes PAGES 2k network working group (http://www.pages-igbp.org/ini/wg/2k-network/intro) and the planning group International Partnership for Ice Core Science, IPICS

(http://www.pages-igbp.org/ini/end-aff/ipics/intro). We thank all of the scientists who made their ice core data available for this study. The Antarctic2k phase 2 database used in this study will be archived at the National Centers for Environmental Information paleoclimatology respository (https://www.ncdc.noaa.gov/data-access/paleoclimatology-data/datasets).

This is a contribution to the PAGES 2k Network [through the Antarctica2k working group]. Past Global Changes (PAGES) is supported by the US and Swiss National Science Foundations. NJA acknowledges funding from the Australian Research

Council through Discovery Project DP140102059. BS acknowledges funding from PNRA (Italian Antarctic Research Programme) IPICS-2kyr-It project. RN is funded by the Swiss NSF (Ambizione grant PZ00P2_154802).

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





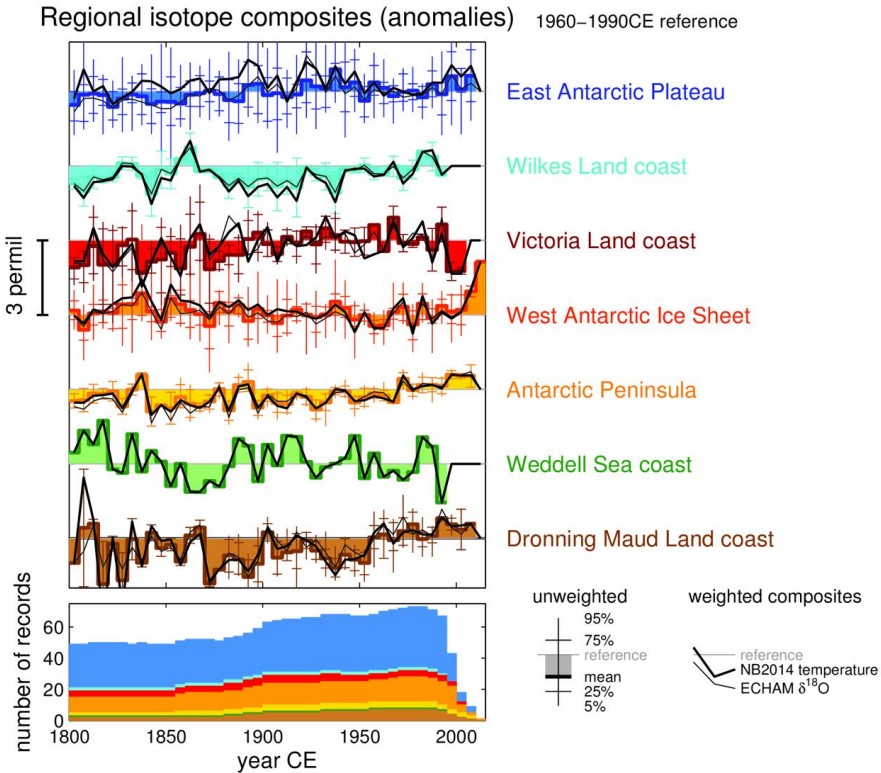

**Figure 2.** Regional $\delta^{18}O$ composite reconstructions over the last 200 years using 5y-binned anomaly data. Both unweighted composites and weighted composites (using both NB2014 temperature and ECHAM $\delta^{18}O$ weighting methods) are shown. For each 5y-bin of the unweighted data, the mean $\delta^{18}O$ anomaly across all records in the climatic region is calculated, as well as the distribution of $\delta^{18}O$ anomalies within each bin. All anomalies are expressed relative to the 1960-1990CE interval. The number of records that contribute to the reconstructions for each region are displayed in the lower panel.



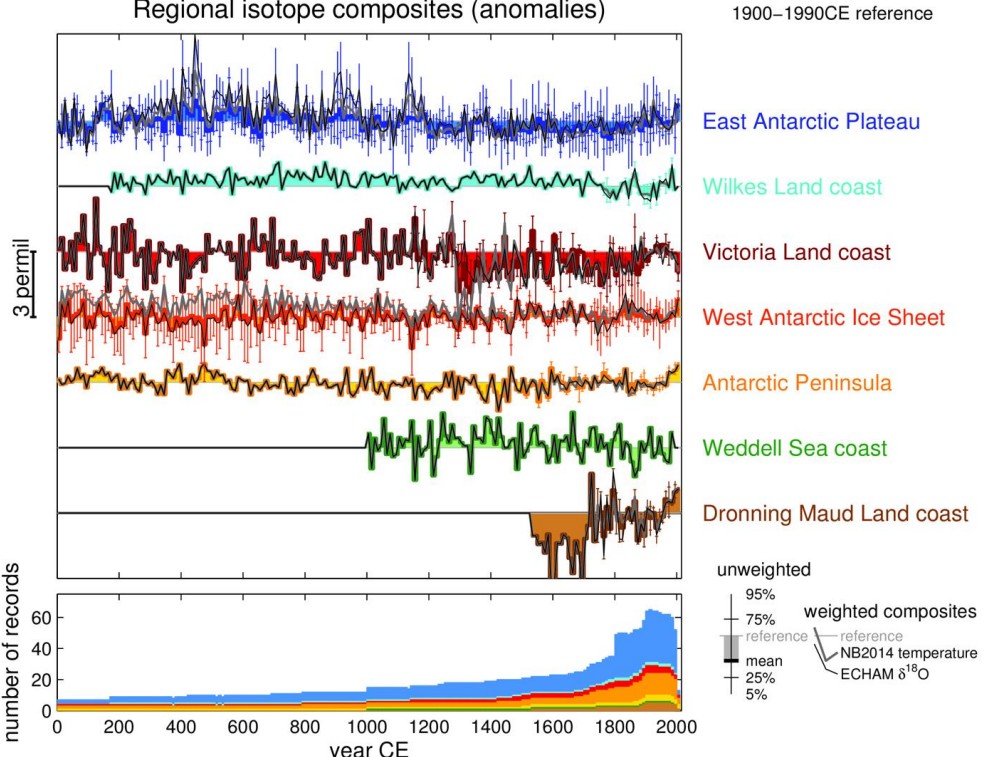

**Figure 3.** Regional $\delta^{18}$O composite reconstructions over the last 2000 years using 10y-binned anomaly data. Both unweighted composites and weighted composites (using both NB2014 temperature and ECHAM $\delta^{18}$O weighting methods) are shown. For each 10y-bin of the unweighted data, the mean $\delta^{18}$O anomaly across all records in the climatic region is calculated, as well as the distribution of $\delta^{18}$O anomalies within each bin. All anomalies are expressed relative to the 1900-1990CE interval. The number of records that contribute to the reconstructions for each region are displayed in the lower panel.

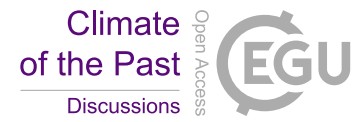

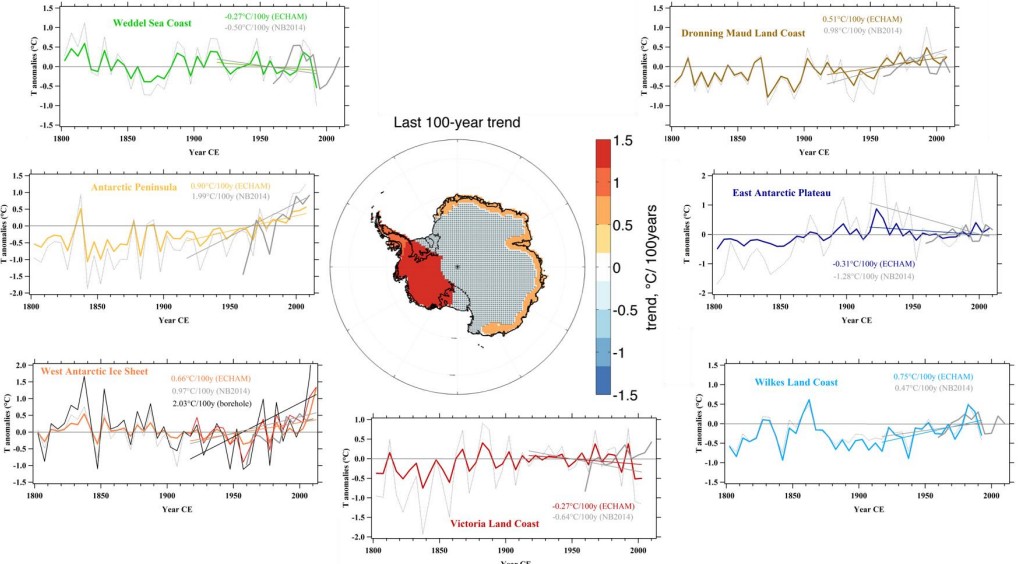

**Figure 4.** Regional temperature (T anomalies in °C, referenced to the 1960-1990 CE interval) reconstructions using 5y-binned data for the past 200y. Weighing method based on the correlation between site T and regional T from NB2014. Temperature scaling method: i) based on the correlation between annual mean regional $\delta^{18}$O and regional T from ECHAM5-wiso forced by ERA-Interim (coloured lines), ii) scaled on NB2014 target over 1960-2000CE (dotted grey lines), iii) West Antarctic Ice Sheet region adjusted to match the temperature trend between 1000 and 1600 CE based on borehole temperature measurements (black line, Orsi et al., 2012). Linear trends are calculated over the last 100 years of the reconstructions (colours match associated reconstruction methods). The map at the centre reports regional trends over the last 100y trend using 5y data based on the ECHAM method adjusted for the West Antarctic Ice Sheet region to borehole data. Hatched areas are not significant (p>0.05).



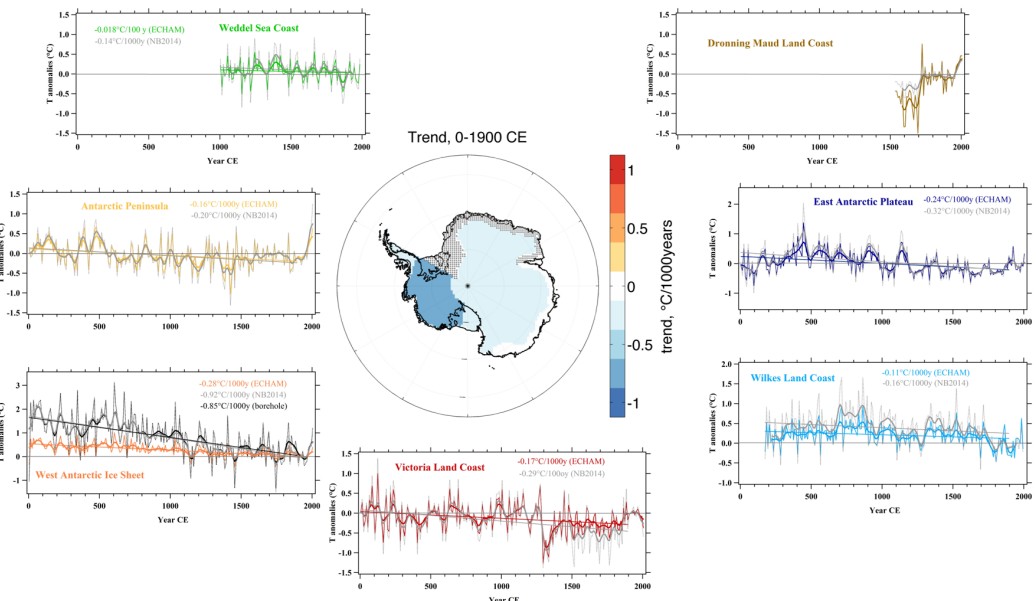

**Figure 5.** Regional temperature (T anomalies in °C, referenced to the 1900-1990 CE period) reconstructions using 10y data for the past 2000y. Weighing method based on the correlation between site T and regional T from NB2014 forced by ERA-Interim. Temperature scaling method: i) based on the correlation between annual mean regional $\delta^{18}O$ and regional T from ECHAM5-wiso forced by ERA-Interim (coloured lines), ii) scaled on NB2014 target over 1960-2000CE (dotted grey lines), iii) West Antarctic Ice Sheet region adjusted to match the temperature trend between 1000 and 1600 CE based on borehole temperature measurements (black line, Orsi et al., 2012). The bold lines are 50y smoothing. Linear trends are calculated over the period 0-1900 CE using 10y binned data. The map at the centre reports the trend values calculated between 0 and 1900 CE using 10y data based on the ECHAM method adjusted for the West Antarctic Ice Sheet region to borehole data (the hatched area is not significant).



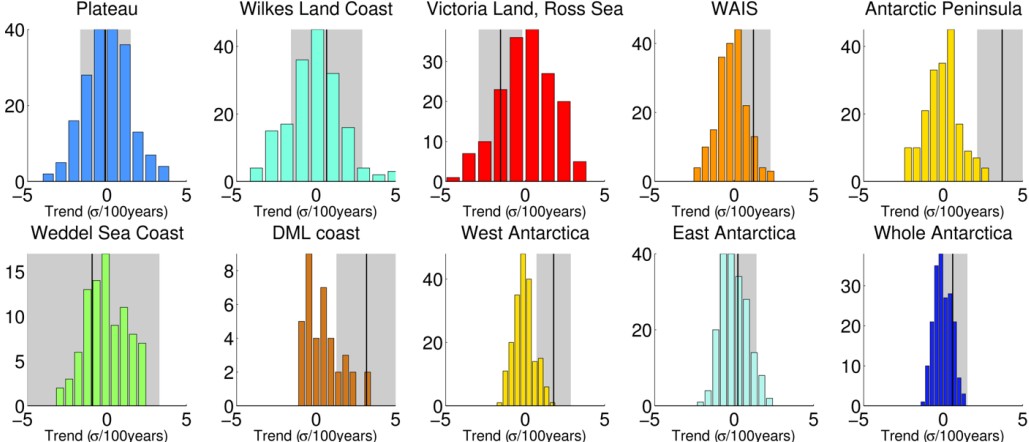

**Figure 6.** Histograms showing the distributions of all 100y trends on normalised, weighted composites over the last 2000 years. Distributions are shown for each climatic region, as well as for East, West and Whole Antarctica composites, and are calculated on 10-year binned composites. The solid vertical lines represent the most recent 100y trend in each reconstruction, and grey shading corresponds to the 5-95% uncertainty range of the last 100y trends. Only for the Antarctic Peninsula does the most recent 100-year trend stand out as unusual compared to the natural range of century-scale trends over the last 2000 years.

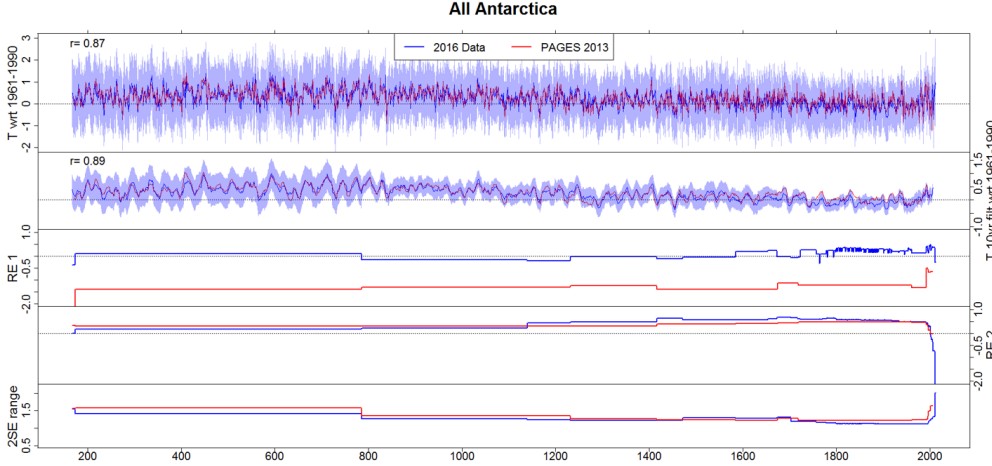

**Figure 7.** Comparison of CPS reconstructions of Antarctic mean temperatures over 167-2010 CE. Red: results from PAGES 2k Consortium (2013). Blue: Updated results using the new ice core isotope data collection described herein and the NB2014 temperature target. Blue shading: 2SE uncertainties of the updated reconstruction. Top panel: Unfiltered interannual reconstructions. Second panel: 10-year running mean of reconstructions. Third (fourth) panel Reduction of Error skill from a split-calibration-verification exercise using 1961-1976 for calibration (verification) and 1977-1991 for verification (calibration). Bottom panel: 2SE reconstruction uncertainty range.





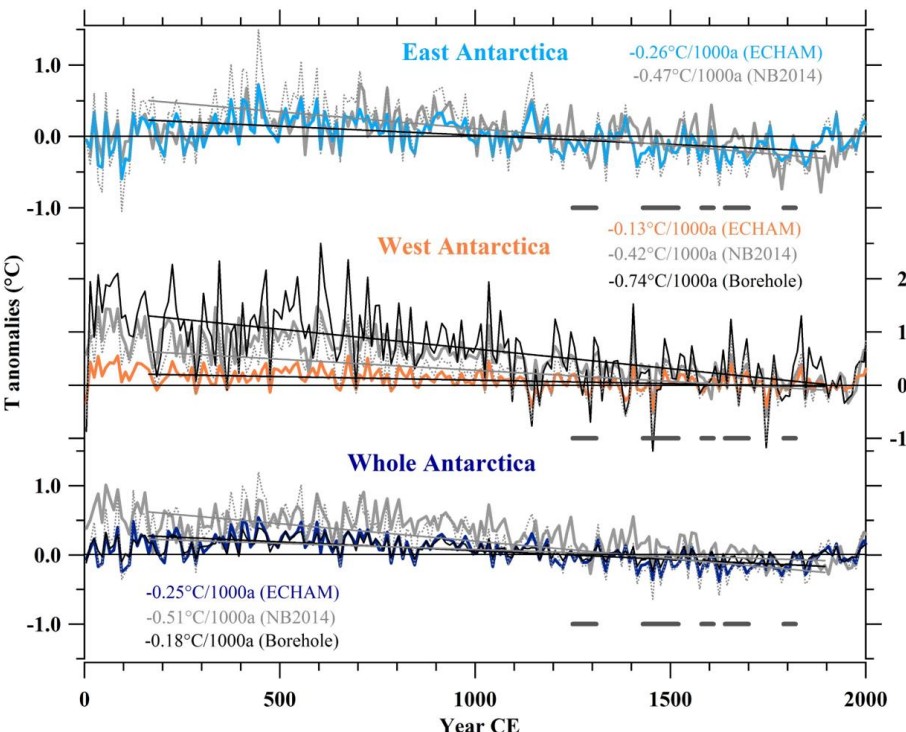

**Figure 8.** Composite temperature reconstructions (T anomalies in °C) for East, West and Whole Antarctica using 10y averages and the different temperature scaling approaches: bold grey lines for the CPS method (2013 method applied to the new database); dotted grey lines for the method which uses the NB2014 variance for scaling (different weighting method compared to CPS); colored lines for the method based on the correlation between annual mean regional $\delta^{18}$O and regional T from ECHAM5-wiso forced by ERA-Interim; black lines for the method adjusted to match the temperature trend between 1000 and 1600 CE based on borehole temperature measurements at WAIS divide (Orsi et al., 2012). Linear trends are calculated between 165 and 1900 CE. The grey horizontal segments correspond to volcanic-solar downturns intervals as defined in PAGES2k Consortium (2013) and corresponding to the following periods: 1251-1310 CE, 1431-1520 CE, 1581-1610 CE, 1641-1700 CE and 1791-1820 CE. Note that the vertical scale for West Antarctica is larger than those for East and Whole Antarctica.



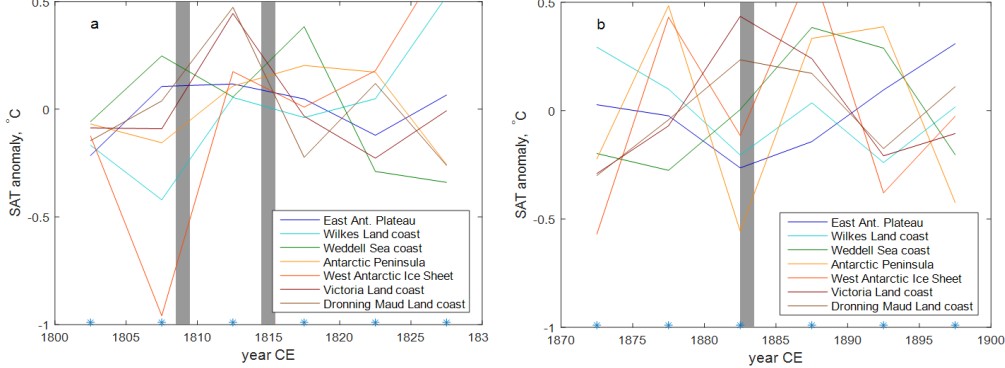

**Figure 9.** Reconstructed regional surface atmospheric temperature anomalies during the periods of 1802.5-1827.5CE (a), and 1872.5-1897.5CE (b) overlapping with the three major tropical volcanic eruptions of 1808/09 and Tambora 1815, and Krakatoa (1883). The reconstruction segments for each region were centred on the means of the corresponding intervals. Stars mark the bin centres of the reconstructions, and the years of the eruptions are highlighted in grey.