# Peer review of "Antarctic climate variability at regional and continental scales over the last 2,000 years"

_Climate of the Past, 2017_

## Referee Comment (RC1) · Anonymous Referee #1 · 12 May 2017

General comments: This MS presented a new Antarctic climate data over the past 2k. Many ice core data compiled under PAGES 2k were evaluated quantitatively as changes in air temperatures. The updated results showed over all cooling trend from 0 to 1900CE and the absence of continental-scale warming over the past 100 yr. Although these results are essentially very similar to those published in previous report (PAGES 2k 2013), this MS is suitable to be published in CP because of its robustness supported by the increased numbers of data and various temperature reconstruction methods.

Specific comments: In this MS, a trend is defined as significant when $p < 0.05$ (e.g., Table 2). Of course, this is widespread practice. However, it should be carefully considered for the choice of this arbitrary-chosen threshold value because significance of the slope is a central issue of this MS. Is there no rule about statistical terms for PAGES

2k consortium? For example, in IPCC AR5, "very likely" indicates 90-100% probability and "likely" indicates 66-100% probability. In this paper, the statistical significance can be ranked such as: ** ($p < 0.05$) and * ($p < 0.10$).

Technical corrections: Conclusions and implications; I feel the conclusions would be too long.

Page 2, l. 29-30; Please add appropriate citations for each phenomenon that may produce non-climate signals.

Page 2, l. 35, "deuterium" > hydrogen?

Page 4, l. 15, "RACMO2.4 model results"; Please add citation for this model results.

Section 3.4.2 and 3.4.3, The final sentences of these sections (We refer to. . ..) may be removed or be moved to the last sentence of the previous paragraph.

Figure 2; NB2014 records (bold black lines) in Wilkes Land and Weddell Sea appears to be horizontal zero lines from 1990-2000. Are these correct? They lines should be shown in gray, if these are reference lines.

Figure 3; Same as Figure 2. NB2014 records (bold black lines) in Wilkes Land, Weddell Sea, and DML appears to be zero for older part of the records. These should be shown in gray.

Figure 9a; In the x-axis, year (1830?) was labelled wrongly (183).

---

## Short Comment (SC1) · 16 May 2017

The PAGES Data Stewardship Integrative Activity seeks to advance best practices for sharing data generated and assembled as part of all PAGES-related activities. As part of this activity, a team of reviewers has been constituted for the "Climate of the Past 2000 years" Special Issue. The data team is reviewing the data handling within each of the CP-Discussion papers in relation to the CP data policy and current best practices. The team has identified essential and recommended additions for each paper, with the goal of achieving a high and consistent level of data stewardship across the 2k Special Issue. We recognize that an additional effort will likely be required to meet the high level of data stewardship envisaged, and we appreciate the dedication and contribution of the authors. This includes the use of Data Citations (see example in supplement). We ask authors to respond to our comments as part of the regular

open interactive discussion. If you have any questions about PAGES Data Stewardship principles, please contact any of us directly.

Best wishes for the success of your paper,

2k Special Issue Data Review Team (Darrell Kaufman, Nerilie Abram, Belen Martrat, Raphael Neukom, Scott St. George) and ex-officio team members (Marie-France Loutre, Lucien von Gunten)

Essential additions for this paper:

(1) Add a "data availability" section that describes where the data can be accessed, including a Data Citation for the temperature reconstructions generated in this study (#4 below), and the target datasets.

(2) Add Data Citations (in addition to publication citations) for all 112 datasets listed in Table S1. For those data not already in a public repository, submit essential metadata along with the time series and include the Data Citation in Table S1.

(3) For those records with previous PAGES 2k IDs, please include cross references to those IDs. These can be found in Table 1 of PAGES2k Consortium (in press). If the data are included in the Iso2k dataset, then please include a cross reference to that data product.

(4) Submit the primary outcome of the data analyses to a public repository and include the Data Citation in "data availability". This includes the (a) composite isotope time series by region (Figs 2 and 3), (b) climate-region temperature reconstructions with the different scaling and binning (Figs 4 and 5), (c) Antarctica-wide (Fig 7) and broad-regions reconstructions (Fig 8) with the different scaling and including the uncertainty statistics.

Recommended elements are:

(5) Archive the weightings used for each of the records in the weighted composite.

(6) Archive the target datasets, including the modeled data.

(7) Archive the results of the trend analyses (Fig. 6).

Please also note the supplement to this comment:
http://www.clim-past-discuss.net/cp-2017-40/cp-2017-40-SC1-supplement.pdf

---

## Referee Comment (RC2) · Anonymous Referee #2 · 1 Jun 2017

General Comments:

This is an important piece of work on temperature variability and change over the Antarctic from large regional areas to continental scales for up to 2000 years. It is based on assembling 112 water stable isotope time series from ice cores and converting these concentrations into temperature using three approaches. And it is a very important data set for testing the performance of coupled climate models, as stated. I have two main issues that should be addressed in the revised manuscript.

First, there is the conversion of isotope variations into temperature. The introduction lays out the many complicating factors contributing to controls on the proxy temperature derived from isotopes. I expected to see in Section 5 an evaluation of the impact of these factors on the results. Two aspects come particularly to mind. Isn't it true

that low altitude ice cores are particularly susceptible to local influences like sea ice variations? (Masson-Delmotte et al. (2008) mention different moisture sources and transportation paths below and above 2000 m elevation). You have many near coastal ice cores shown in Figure 1. How do such effects contaminate/modulate your results? The other issue is that the stable isotope concentration is a precipitation signal (you use precipitation weighting with the ECHAM5-wiso simulated isotope results – good) and a better temperature is the condensation temperature where the precipitation forms, not the surface temperature. I didn't see any mention of this, so what do your derived temperatures actually represent? Changes to advection conditions (and thus the moisture source) can and do happen and influence the condensation temperature, and the advection conditions may not stay fixed across decades to millennia. Consider the following paper: Gorodetskaya, I. V., M. Tsukernik, K.Claes, M.F.Ralph, W.D.Neff,and N. P. M. Van Lipzig (2014), The role of atmospheric rivers in anomalous snow accumulation in East Antarctica, Geophys. Res. Lett.,41, 6199-6206 ,doi:10.1002/2014GL060881. Such atmospheric river events contribute up to 80% of the annual accumulation at Princess Elizabeth Station in East Antarctica, with significant interannual variability; in some cases the precipitation origin resides in the subtropics. This perspective surely adds to the envelope of uncertainty for the temperature reconstructions and it seems unlikely that ECHAM5 even at 100 km resolution would adequately capture such narrow moisture transport events.

The second major issue is that section 4.3 on "Response to volcanic forcing" comes across as so premature that it should be dropped from the manuscript. There is no consistent signal in Figure 9. Why should one believe the results for some regions and not others without some compelling reason other than this is the expected result. Page 18, line 18 calls this "only a preliminary attempt to assess the possible climatic response of Antarctic climate to short-term volcanic eruptions". Page 19, line 4 says "showed possible negative temperature effects in some regions". The big story here is the temperature reconstructions that need substantially more consideration as outlined above. The inconclusive volcanic aspect is an unnecessary distraction. Better to leave

this topic for detailed exploration elsewhere.

Lesser aspects:

1. Page 2, lines 17 and 19: please discriminate between the two Jones et al. (2016) references.

2. "Weddel" is incorrectly used in Tables 3-4 and Figures 1 and 4-6.

3. Consider redoing the time series in Figures 4 and 5 that are too faint.

4. What is the source of the NB2014 data used extensively in this analysis?

5. Page 9, line 31 onward to Page 10, line 10 discusses the very variable delta 18O-temperature slopes. To me this says that the conversion from isotope concentrations to temperature is not robust except perhaps in the Antarctic interior. This is another reason for having an extended evaluation of isotope-temperature conversion at the end of your manuscript and how the associated uncertainties impact your results. This information (basically the error bars) is needed to ensure that a robust evaluation of climate model results can be conducted.

---

## Author Response (AR1)

**Authors' response:**

We thank the reviewers for their constructive comments.

Below, the reviewers' comments are in black, our responses to the individual comments are displayed in red and the modified text is in blue.

**Anonymous Referee #1**

General comments: This MS presented a new Antarctic climate data over the past 2k. Many ice core data compiled under PAGES 2k were evaluated quantitatively as changes in air temperatures. The updated results showed over all cooling trend from 0 to 1900CE and the absence of continental-scale warming over the past 100 yr. Although these results are essentially very similar to those published in previous report (PAGES 2k 2013), this MS is suitable to be published in CP because of its robustness supported by the increased numbers of data and various temperature reconstruction methods.

We thank the referee for her/his positive comments.

Specific comments: In this MS, a trend is defined as significant when p < 0.05 (e.g., Table 2). Of course, this is widespread practice. However, it should be carefully considered for the choice of this arbitrary-chosen threshold value because significance of the slope is a central issue of this MS. Is there no rule about statistical terms for PAGES 2k consortium? For example, in IPCC AR5, "very likely" indicates 90-100% probability and "likely" indicates 66-100% probability. In this paper, the statistical significance can be ranked such as: \*\* (p < 0.05) and \* (p < 0.10).

There is no rule for statistical terms within the PAGES 2k consortium (the community guidelines are more around data stewardship). But the reviewers' suggestion can easily be added to our tables to highlight significance at the p < 0.10 level as well as at the more stringent p < 0.05 level. This revision does not alter any of the interpretations made in the manuscript.

We provide in table 2, 3 and 4 the p-value of the trends, so that the reader can use the significance level of his choice.

Technical corrections: Conclusions and implications; I feel the conclusions would be too long.

We have shortened them. The paragraph between lines 22 and 34 of page 18 (original manuscript) has been reduced. At page 19 the lines between 2 and 7 have been deleted and the lines between 27 and 33 have been moved to section 4.1.4.

Page 2, I. 29-30; Please add appropriate citations for each phenomenon that may produce nonclimate signals.

We have added the following citations: intermittency of precipitation (Masson-Delmotte et al., 2011: Clim. Past, 7, 397–423, doi:10.5194/cp-7-397-2011; Sime et al., 2009: GRL, doi:10.1029/2009GL038982); precipitation source region variability (Sodemann and Stohl, 2009: GRL, 36, L22803, doi:10.1029/2009GL040242); post depositional effects of snow layers including wind drift and scouring, sublimation, and snow metamorphism (Frezzotti et al., 2007: JGR, 112,

F02032, doi:10.1029/2006JF000638; Ekaykin et al., 2014: Ann. Glaciol., 55, 259-266, doi: 10.3189/201AoG66A189; Touzeau et al., 2016: The Cryosphere, 10, 837–852, doi:10.5194/tc-10-837-2016; Hoshina et al., 2014: J. Geophys. Res., 119, 274–283; Steen-Larsen et al., 2014: Clim. Past, 10, 377–392, doi:10.5194/cp-10-377-2014).

Page 2, I. 35, "deuterium" > hydrogen?

Done.

Page 4, I. 15, "RACMO2.4 model results"; Please add citation for this model results.

These model results are shown in the companion paper on snow accumulation rate by Thomas et al. (under review for this CP special issue): their figures 2 and 3. We refer to this paper in the text, as well as to the reference Van Wessem et al., 2014 (J. Glaciol., 60, 761-770), for the model.

Section 3.4.2 and 3.4.3, The final sentences of these sections (We refer to. . ..) may be removed or be moved to the last sentence of the previous paragraph.

We have moved these sentences, as well as the one for ECHAM, at the end of 3.4 Paragraph (Page 9).

Figure 2; NB2014 records (bold black lines) in Wilkes Land and Weddell Sea appears to be horizontal zero lines from 1990-2000. Are these correct? They lines should be shown in gray, if these are reference lines.

There are no data in correspondence to these features. We have modified this figure.

Figure 3; Same as Figure 2. NB2014 records (bold black lines) in Wilkes Land, Weddell Sea, and DML appears to be zero for older part of the records. These should be shown in gray.

Same as above.

Figure 9a; In the x-axis, year (1830?) was labelled wrongly (183).

We have modified the figure.

**Anonymous Referee #2**

General Comments:

This is an important piece of work on temperature variability and change over the Antarctic from large regional areas to continental scales for up to 2000 years. It is based on assembling 112 water stable isotope time series from ice cores and converting these concentrations into temperature

using three approaches. And it is a very important data set for testing the performance of coupled climate models, as stated.

We thank the referee for her/his positive comments.

I have two main issues that should be addressed in the revised manuscript.

First, there is the conversion of isotope variations into temperature. The introduction lays out the many complicating factors contributing to controls on the proxy temperature derived from isotopes. I expected to see in Section 5 an evaluation of the impact of these factors on the results.

Our introduction summarizes the different processes other than surface temperature which may affect isotope variations, as described at page 2 lines 28–31 and page 2 line 40, page 3 line 13, e.g. changes of the origin of moisture sources, intermittency in precipitation, snow drift, snow-air exchanges, snow metamorphism, diffusion in ice cores, as well as boundary layer processes which can modify the relationship between condensation temperature and surface air temperature.

Our analysis is not focused on the understanding of these processes but on the assessment of relationships between ice core isotopic composition and surface air temperature, as a focus of PAGES2k reconstructions in various regions. For this purpose, we work both on statistical analyses of regional temperature reconstructions and ice core records, as well as the coherent framework provided by the ECHAM5-wiso model, where we also investigate the relationships between surface air temperature and precipitation isotopic composition.

The conclusions that isotope-temperature relationships differ from region to region indeed call for further investigations to understand which of the processes identified above differ from region to region and explain differences in the relationships.

This further analysis is beyond feasibility at this stage. Indeed, recent studies showed a relationship between the range values of isotopic composition in precipitation or shallow cores and air mass pathways, but most studies were restricted to specific arrival points (e.g. Markle et al., J. Geophys. Res., 2012; Sinclair et al. Geoph. Res. Lett., 2014 Kurita et al., Geoph. Res. Lett., 2016; Ramahan et al., The Holocene, 2016; Schlosser et al., The Cryosphere, 2017). Systematic analyses of moisture transport pathways and evaporation conditions through time and space have yet to be performed and combined with isotopic records; such analysis is beyond the scope of our manuscript.

Post-deposition effects cannot be evaluated neither from ice core data, nor in the ECHAM5-wiso model (as they are not resolved and we rely on precipitation outputs); obtaining new monitoring data in multiple Antarctic regions is crucial for an improved understanding, when the implementation of post-deposition processes such as snow-air exchanges is in progress in snow models (Touzeau et al., in prep.).

We discuss the limits of making a linear isotope/temperature relationship in section 3.4. We are clear in the manuscript about the fact that the temperature scaling is approximate, and we make an attempt at evaluating the uncertainty associated with it by using different approaches, and independent datasets for the calibration. To our knowledge, we have been exhaustive in the range of approaches possible. The detailed study of the impact of each fractionation process is impossible, simply because we do not have independent observations to constrain them individually. This is a whole field of research that cannot be addressed in a single community paper. Here, our approach

was to do our best at reconstructing temperature based on water isotopes. Surface temperature is our objective, and we use isotopes to infer surface temperature. We look at all processes in "bulk", acknowledging that several mechanisms responsible for isotope fractionation may be correlated, and reinforce each other in a simple regression.

Finally, most of our results are based on looking at the  $\delta^{18}$ O anomalies, independently of the temperature scaling, and our conclusions about the significance of the trends hold independently of the temperature scaling.

We added a description of the method at the beginning of section 3.4:

The relationship between  $\delta^{18}$ O and local surface temperature is complicated by the influence of a large range of processes (origin of moisture sources, intermittency in precipitation, snow drift, snow-air exchanges, snow metamorphism, diffusion in ice cores). It is not possible to consider each process independently, because in many cases, there are simply no observations to constrain them well enough. However, the atmospheric circulation often leads to several processes to be correlated (reduced sea ice, increased precipitation and warmer temperature, for instance). We follow here the classical approach, which is to perform a linear regression of ice core  $\delta^{18}$ O with local surface temperature, on the regional average products. This method has the advantage of looking at all the climatic processes influencing  $\delta^{18}$ O in "bulk", and the use of regional average allows us to limit the influence of small scale processes.

Two aspects come particularly to mind. Isn't it true that low altitude ice cores are particularly susceptible to local influences like sea ice variations? (Masson-Delmotte et al. (2008) mention different moisture sources and transportation paths below and above 2000 m elevation). You have many near coastal ice cores shown in Figure 1. How do such effects contaminate/modulate your results?

In our paper, we indeed selected seven distinct climatic regions: the Antarctic Peninsula, the West Antarctic Ice Sheet, the East Antarctic Plateau, and four coastal domains of East Antarctica, using the 2,000 m elevation contour. The reason for choosing this 2,000 m elevation as a boundary between coastal/low altitude regions and the East Antarctic Plateau is based on the reasons mentioned by the reviewer and the work presented in Masson-Delmotte et al ., 2008. Our regional selections are also supported by spatial correlation of temperature using the NB2014 reanalysis data and by regional atmospheric RACMO2.4 model results (surface mass balance and temperature spatial correlation plots). In the region selection paragraph 3.1 we have better explained the possible different moisture sources and transportation paths below and above 2000 m elevation referring to Masson-Delmotte et al. (2008) paper. Considerable differences between cores from the plateau and the coast have been reported in the Altnau et al. 2015 (The Cryosphere) for the DML area covering the most recent period and where several records are present. We will refer to this paper too.

**Addition to section 3.1:**

In particular, we separated coastal regions (below 2000m altitude) from the East Antarctic Plateau: coastal sites receive moisture from the high-latitude Southern Ocean, and are affected by the nearby sea ice cover (Masson-Delmotte et al., 2008). In contrast, high altitude sites receive moisture

that has travelled at higher altitude, originating from further afield, and from clear sky precipitation (Ekaykin et al., 2004).

The other issue is that the stable isotope concentration is a precipitation signal (you use precipitation weighting with the ECHAM5-wiso simulated isotope results – good) and a better temperature is the condensation temperature where the precipitation forms, not the surface temperature. I didn't see any mention of this, so what do your derived temperatures actually represent?

The focus of this PAGES2K effort is a reconstruction of regional surface air temperature, and therefore this is our target.

Historically, a linear relationship was found between local surface temperature and the isotopic composition of surface snow along spatial traverse data (e.g. Dansgaard et al., Nature, 1975; Jouzel et al., J. Geophys. Res., 1983; Morgan et al., Clim. Change, 1985, Aristarain et al., Geoph. Res. Lett., 1986 and Rozanski et al., Science, 1992). These results lead to the common reconstruction of surface temperature from the isotopic composition recorded in ice cores using a single spatial relationship. Our study goes beyond this classical approach in assessing different options for the isotope-temperature scaling.

Moreover, Stenni et al. (2016) recently reported a better correlation between Dome C precipitation  $\delta^{18}$ O and 2 m air temperature than for inversion temperature, whatever the considered timescale (daily, monthly or inter-annual scale). This calls for more assessment of the driver of final isotopic depletion in central Antarctica. Finally, the ECHAM5-wiso framework offers in principle the possibility to analyse such relationships; however, condensation temperature was not archived as a standard diagnostic from daily model outputs, precluding further analyses.

We added a few sentences clarifying the target of our reconstruction at the start of section 3.4:

We follow here the classical approach, which is to perform a linear regression of ice core  $\delta^{18}$ O with local surface temperature, on the regional average products. This method has the advantage of looking at all the climatic processes influencing  $\delta^{18}$ O in "bulk", and the use of regional average allows us to limit the influence of small-scale processes.

Changes to advection conditions (and thus the moisture source) can and do happen and influence the condensation temperature, and the advection conditions may not stay fixed across decades to millennia. Consider the following paper: Gorodetskaya, I. V., M. Tsukernik, K.Claes, M.F.Ralph, W.D.Neff, and N. P. M. Van Lipzig (2014), The role of atmospheric rivers in anomalous snow accumulation in East Antarctica, Geophys. Res. Lett., 41, 6199-6206 ,doi:10.1002/2014GL060881. Such atmospheric river events contribute up to 80% of the annual accumulation at Princess Elizabeth Station in East Antarctica, with significant interannual variability; in some cases the precipitation origin resides in the subtropics. This perspective surely adds to the envelope of uncertainty for the temperature reconstructions and it seems unlikely that ECHAM5 even at 100 km resolution would adequately capture such narrow moisture transport events.

This study is of course limited by the spatial resolution of the model (here run at T106 resolution). At such a resolution, the simplified mapping of costal topographies affects the atmospheric

circulation on these areas, and processes such as uplift of air masses onto the continent or katabatic winds.

We have not performed systematic assessments of the skills of ECHAM5-wiso for overall moisture transport towards Antarctica, but ECHAM5-wiso is used here nudged to ERA-interim outputs. A recent study using radiosounding data has shown the good skills of ERA-interim for overall moisture transport towards Antarctica, including the representation of extra-tropical storms (PhD thesis, Ambroise Dufour).

In Antarctica, recent atmospheric river events have been diagnosed in the Atlantic sector, in the last years. A large atmospheric river event was also identified in July 2012 towards and above the Greenland ice sheet, where isotopic measurements were available. Bonne et al. (2014) has demonstrated the ECHAM5-wiso skills to capture both this atmospheric river and the associated water vapour isotopic composition. While not in an Antarctic context, this finding suggests that our model framework is able to correctly simulate such processes. Further analyses would be needed to identify such events systematically around Antarctica since 1979 and understand how they may affect the isotopic signal and the resulting isotope-temperature relationships at inter-annual scale.

Moreover, in our reconstructions using 5 and 10 year average values we should in part overcome this interannual variability of precipitation and surface mass balance.

We added a sentence to section 2.3 to highlight the skill of ECHAM5-wiso to capture atmospheric rivers:

A study of the 2012 atmospheric river event in Greenland has demonstrated the skill of ECHAM5wiso to reproduce these events, with a good representation of the water isotope signature (Bonne et al., 2014).

The second major issue is that section 4.3 on "Response to volcanic forcing" comes across as so premature that it should be dropped from the manuscript. There is no consistent signal in Figure 9. Why should one believe the results for some regions and not others without some compelling reason other than this is the expected result. Page 18, line 18 calls this "only a preliminary attempt to assess the possible climatic response of Antarctic climate to short-term volcanic eruptions". Page 19, line 4 says "showed possible negative temperature effects in some regions". The big story here is the temperature reconstructions that need substantially more consideration as outlined above. The inconclusive volcanic aspect is an unnecessary distraction. Better to leave this topic for detailed exploration elsewhere.

We thank the reviewer for this comment and can partly agree that the analysis may not be a 100% match to a general content of the manuscript. This section indeed shows in brief what the compiled dataset can further be used for beyond the presented analysis for the trends, touching on a much more extensive issue of climatic drivers throughout the late Holocene. Regardless of what actually drives the late Holocene cooling (so called Neoglaciation) - volcanic, orbital forcing, circulation changes triggered by volcanic/solar forcing or a combination of these factors, the study of the climate response at a short, sub-decadal timescale, contributes to answering the issue of a direct regional climate sensitivity. The major motivation for us to retain this section as is, is that it clearly demonstrates the limitations of the newly compiled dataset. In our opinion this should be considered as important as reporting the "positive" results in the rest of the manuscript.

We further demonstrate that at the present quality level of the database, unambiguous research on this topic should use the chronology control/reassessment to promote a better synchronization of the records at least on the regional level as a starting point. Emphasizing this in the present manuscript in our opinion gives a future prospective research on the topic a good starting point.

Lesser aspects:

1. Page 2, lines 17 and 19: please discriminate between the two Jones et al. (2016) references. It is the same reference. We have deleted one citation.

2. "Weddel" is incorrectly used in Tables 3-4 and Figures 1 and 4-6.

We have corrected this error in labelling of the Tables and Figures.

3. Consider redoing the time series in Figures 4 and 5 that are too faint.

We have modified the figures to fix this issue.

4. What is the source of the NB2014 data used extensively in this analysis?

We added in section 2.2 and in Data Availability section the link to the NB2014 data archive, which is: http://polarmet.osu.edu/datasets/Antarctic\_recon/

5. Page 9, line 31 onward to Page 10, line 10 discusses the very variable delta 18Otemperature slopes. To me this says that the conversion from isotope concentrations to temperature is not robust except perhaps in the Antarctic interior.

We have added Root Mean Square Error associated to each region linear regression, as reported in the table below.

| Geographic region           | slope (°C/‰)    | slope (‰/°C)    | r      | p-value |
|-----------------------------|-----------------|-----------------|--------|---------|
| 1. East Antarctic Plateau   | $0.95 \pm 0.05$ | $1.05 \pm 0.06$ | 0.6249 | 0.0001  |
| 2. Wilkes Land Coast        | $1.91 \pm 0.11$ | $0.52\pm0.03$   | 0.4385 | 0.0084  |
| 3. Weddell Sea Coast        | $1.01\pm0.06$   | $0.99\pm0.06$   | 0.3411 | 0.0449  |
| 4. Antarctic Peninsula      | $2.50\pm0.15$   | $0.40\pm0.02$   | 0.3145 | 0.0658  |
| 5. West Antarctic Ice Sheet | $1.04\pm0.06$   | $0.96\pm0.05$   | 0.5894 | 0.0002  |
| 6. Victoria Land Coast      | $0.83\pm0.05$   | $1.21\pm0.07$   | 0.4916 | 0.0027  |
| 7. Dronning Maud Land Coast | $1.08\pm0.06$   | $0.93\pm0.05$   | 0.3868 | 0.0217  |
| West Antarctica             | $1.03 \pm 0.06$ | $0.97\pm0.05$   | 0.6195 | 0.0001  |
| East Antarctica             | $1.00\pm0.05$   | $1.00\pm0.05$   | 0.5842 | 0.0002  |

| All Antarctica | $1.02\pm0.06$ | $0.98\pm0.05$ | 0.5649 | 0.0004 |
|----------------|---------------|---------------|--------|--------|
|                |               |               |        |        |

There are indeed substantial variations in the isotope-temperature slopes produced by different approaches (model or observational-based) in some parts of Antarctica. Each of the different approaches has advantages and limitations, and it is not possible with the data sources currently available to say which approach is the most robust. We note though that the differences in isotope-temperature conversions only alter the magnitude and significance of the temperature trends produced, but do not alter the sign of the trends. In this regard, we would argue that the ice core isotope data provide a robust representation of changes in Antarctic temperature, but that there is uncertainty in the scaling of isotopic changes to temperature in some regions. We propose to clarify the text to more clearly specify the implication of the differences in isotope-temperature conversions have in specific regions where there are differences in the conversions produced by different methods.

This is another reason for having an extended evaluation of isotope-temperature conversion at the end of your manuscript and how the associated uncertainties impact your results. This information (basically the error bars) is needed to ensure that a robust evaluation of climate model results can be conducted.

In our paper, we provide 4 different temperature reconstruction data sets (see figure 8 and Fig S15 in supplementary materials):

- 1) "ECHAM" method: ECHAM5-wiso to determine the regional  $\delta^{18}$ O temperature relationship;
- 2) "NB2014" method: scales the normalized record to the instrumental period temperature variance;
- 3) "borehole" method: temperature scaling using the borehole measurements at WAIS divide core site (Orsi et al., 2012). This method is used for WAIS region, the West Antarctic Ice Sheet and Whole Antarctica only.
- 4) "CPS" method: used in the previous PAGES 2013 consortium paper and here using NB2014 data as a target.

The quite good agreement among these 4 method supports the robustness of our reconstructions and could provide information on T reconstruction uncertainties at regional and continental scale.

**We discussed the uncertainty in the temperature reconstruction at the start of Section 4.1.2:**

We estimate the uncertainty of the slope based on the  $\pm 2\sigma$  uncertainty in the regression parameters. The robustness of the slope estimation to individual data points was further checked by taking out 10% of data points randomly and calculating the slope again, but the uncertainty estimates this yields is much smaller than the uncertainty based on regression parameters. The slopes obtained by each of the temperature scaling methods are presented in Table 3. The uncertainty in the amplitude of the 0-1900 CE trend is dominated by the uncertainty in the temperature scaling of the composite.

We also added in the conclusions:

Similar conclusions are reached using different approaches (ECHAM, NB2014 and borehole methods) giving support to our temperature reconstructions.

**Interactive comment on data stewardship**

The PAGES Data Stewardship Integrative Activity seeks to advance best practices for sharing data generated and assembled as part of all PAGES-related activities. As part of this activity, a team of reviewers has been constituted for the "Climate of the Past 2000 years" Special Issue. The data team is reviewing the data handling within each of the CP-Discussion papers in relation to the CP data policy and current best practices. The team has identified essential and recommended additions for each paper, with the goal of achieving a high and consistent level of data stewardship across the 2k Special Issue. We recognize that an additional effort will likely be required to meet the high level of data stewardship envisaged, and we appreciate the dedication and contribution of the authors. This includes the use of Data Citations (see example in supplement). We ask authors to respond to our comments as part of the regular open interactive discussion. If you have any questions about PAGES Data Stewardship principles, please contact any of us directly.

Best wishes for the success of your paper, 2k Special Issue Data Review Team (Darrell Kaufman, Nerilie Abram, Belen Martrat, Raphael Neukom, Scott St. George) and ex-officio team members (Marie-France Loutre, Lucien von Gunten).

As soon as we know that the paper will be accepted for publication we will assemble our temperature reconstruction data to submit to NOAA and assemble metadata and data citations for all records.

Essential additions for this paper:

(1) Add a "data availability" section that describes where the data can be accessed, including a Data Citation for the temperature reconstructions generated in this study (#4 below), and the target datasets.

The data will be uploaded to NOAA and we will generate a data citation for the temperature reconstructions generated in this study.

We have added a "Data availability" section to the revised manuscript:

**Data availability**

The target dataset of NB2014 is archived at: http://polarmet.osu.edu/datasets/Antarctic\_recon/.

The table S1 (Supplementary Information) contains the data citations, as well as the primary bibliographic reference, for each record used in this study. We used the URL links for existing published data from public repositories.

The regional and continental scale isotopic composites and temperature reconstructions generated during the current study are available in the NOAA World Data Center for Paleoclimatology (WDC Paleo) at the following link: https://www.ncdc.noaa.gov/paleo/study/22589.

(2) Add Data Citations (in addition to publication citations) for all 112 datasets listed in Table S1. For those data not already in a public repository, submit essential metadata along with the time series and include the Data Citation in Table S1.

We have added data citations in Table S1 (see above) for all the records used in generating the composites.

(3) For those records with previous PAGES 2k IDs, please include cross references to those IDs. These can be found in Table 1 of PAGES2k Consortium (in press). If the data are included in the Iso2k dataset, then please include a cross reference to that data product.

We have provided in Table S1 full cross referencing for all records used in the previous 2k temperature paper (PAGES 2k Consortium, 2013) and the recently published 2k Scientific Data paper (PAGES 2k Consortium, 2017).

(4) Submit the primary outcome of the data analyses to a public repository and include the Data Citation in "data availability". This includes the (a) composite isotope time series by region (Figs 2 and 3), (b) climate-region temperature reconstructions with the different scaling and binning (Figs 4 and 5), (c) Antarctica-wide (Fig 7) and broadregions reconstructions (Fig 8) with the different scaling and including the uncertainty statistics.

As soon as we know that the paper will be accepted for publication we will archive the different temperature reconstructions to NOAA (see above).

Recommended elements are:

(5) Archive the weightings used for each of the records in the weighted composite.

The matlab codes could be available from the authors upon request.

(6) Archive the target datasets, including the modeled data.

The target dataset of NB2014 is already archived at: http://polarmet.osu.edu/datasets/Antarctic recon/. We have provided this link in the data availability section.

(7) Archive the results of the trend analyses (Fig. 6).

All the trend analysis statistics is reported in Table 3 and 4.

**Antarctic climate variability at regional and continental scales over the last 2,000 years**

Barbara Stenni1,2, Mark A. J. Curran3,4, Nerilie J. Abram5,6, Anais Orsi7, Sentia Goursaud7,8, Valerie Masson-Delmotte7, Raphael Neukom9, Hugues Goosse10, Dmitry Divine11,12, Tas van Ommen3,4, Eric J. Steig13, Daniel A. Dixon14, Elizabeth R. Thomas15, Nancy A. N. Bertler16,17, Elisabeth Isaksson11, Alexey Ekaykin18,19, Massimo Frezzotti20, Martin Werner204, Massimo Frezzotti21

[revised manuscript text omitted]
   | $0.95 \pm 0.05 + 0.05 + 0.06$                | $\underline{1.05\pm0.06}$   | 0.62        | 0.0001 <0.001  |  |
| 2. Wilkes Land Coast        | $1.91 \pm 0.11 + 0.23$                       | $\underline{0.52\pm0.03}$   | 0.44        | 0.0084 0.0084  |  |
| 3. Weddell Sea Coast        | $1.01 \pm 0.06 + 0.34$                       | $\underline{0.99 \pm 0.06}$ | 0.34 | 0.0449 0.0449  |  |
| 4. Antarctic Peninsula      | $2.50 \pm 0.15 + 0.13$                       | $\underline{0.40\pm0.02}$   | 0.31        | 0.0658 0.0658  |  |
| 5. West Antarctic Ice Sheet | $\underline{1.04 \pm 0.06} \underline{0.57}$ | $\underline{0.96\pm0.05}$   | 0.59        | 0.0002 <0.001  |  |
| 6. Victoria Land Coast      | $\underline{0.83 \pm 0.05} \underline{0.59}$ | $1.21 \pm 0.07$             | 0.49        | 0.0027 0.00271 |  |
| 7. Dronning Maud Land Coast | $\underline{1.08 \pm 0.06} \underline{0.36}$ | $\underline{0.93 \pm 0.05}$ | 0.39        | 0.0217 0.0271  |  |
| East West Antarctica        | $1.03 \pm 0.06 + 0.58$                       | $0.97 \pm 0.05$             | 0.62        | 0.0001 <0.001  |  |
| West East Antarctica        | $1.00 \pm 0.05 + 0.55$                       | $1.00 \pm 0.05$             | 0.58        | 0.0002 <0.001  |  |
| All Antarctica              | $1.02 \pm 0.06 + 0.60$                       | $\underline{0.98\pm0.05}$   | 0.56        | 0.0004 <0.001  |  |

10

15

5

[revised manuscript text omitted]
 - | -0.006 (0.118)           | -0.006 (0.101)           | 100 (12.76)             | -0.054 (0.220)          | -0.074 (0.102)          | 99.9 (13.6)                      |  |  |
| Ross Sea                               | 1                        |                          |                         |                         |                         |                                  |  |  |
| 7. Dronning Maud Land                  | -0.032 (0.366)           | -0.027 (0.448)           | 95.3 (0.1)              | 0.147 (0.000)           | 0.158 (0.001)           | 100 (99.8)                       |  |  |
| Coast                           |                          |                          |                         |                         |                         |                                  |  |  |
| East-West Antarctica                   | 0.000 (0.437)-           | -0.001 (0.118)-          | 95.57            | 0.054            | 0.082            | 100 (70 8)00 8                   |  |  |
| 1                                      | 0.000 (0.+57)            | -0.001 (0.110)    | (11.2) 100       | (0.021) 0.037    | (0.001) 0.064    |                                  |  |  |
| 1                                      | <del>0.003 (0.000)</del> | <del>0.002 (0.000)</del> | <del>(100)</del>        | <del>(0.035)</del>      | <del>(0.003)</del>      | (55.1)                           |  |  |
| West-East Antarctica                   | -0.003            | 0.002 (0.000)-           | 100 (100)05 57          | 0.037            | 0.064            | 00.9 (55.1)100                   |  |  |
| 1                                      | (0.000) 0.000     | -0.002 (0.000) -  | $\frac{100(100)}{2007}$ | (0.035) 0.054    | (0.003) 0.082    | 99.0 (55.1)100 |  |  |
| 1                                      | <del>(0.437)</del>       | <del>0.001 (0.118)</del> | <del>(11.2)</del>       | <del>(0.021)</del>      | <del>(0.001)</del>      | <del>(/У.ð)</del>         |  |  |
| All Antarctica                         | -0.002 (0.000)           | -0.002 (0.000)           | 100 (99.5)              | 0.044 (0.005)           | 0.073 (0.000)           | 99.9 (76.2)                      |  |  |

10

**4.1.2 Long term trends in weighted reconstructions**

[revised manuscript text omitted]

|                             | start | end         | ECH                        | AM                | NB20                | 14         | borehole            | Bboreho
le               |
|-----------------------------|--------------|-------------|----------------------------|-------------------|----------------------------|-------------------|----------------------------|-----------------------------|
| Geographic region    | date  | date | Slope
(°C/1000y) | p-value    | Slope
(°C/1000y) | p-value    | Slope
(°C/1000y) | p-value              |
| 1. East Antarctic Plateau   | 0     | 1900 | -0.38±0.14          | <0.0001 | -0.32±0.12          | <0.0001 | NaN                 | NaN                         |
| 2. Wilkes Land Coast        | 170   | 1900 | -0.24±0.17                 | 0.0072            | -0.16±0.13                 | 0.0161     | NaN                 | NaN                         |
| 3. Weddell Sea Coast        | 1000  | 1900 | -0.24±0.54          | 0.3763     | -0.12±0.27          | 0.3763     | NaN                 | NaN                         |
| 4. Antarctic Peninsula      | 0     | 1900 | -0.52±0.23          | <0.0001 | -0.20±0.09                 | <0.0001 | NaN                 | NaN                         |
| 5. West Antarctic Ice Sheet | 0     | 1900 | -0.47±0.10          | <0.0001 | -0.92±0.16                 | <0.0001 | -0.55±0.11                 | <0.0001           |
| 6. Victoria Land Coast      | 0     | 1900 | $\frac{-0.34}{\pm 0.18}$   | 0.0003     | -0.29±0.12                 | <0.0001 | NaN                        | 0.0003N
aN |
| 7. Dronning Maud Land Coast | 1530  | 1900 | 5.96±3.27                  | 0.0007     | 0.97±0.62           | 0.0032     | NaN                        | 0.0007N
aN |
| West Antarctica             | 0            | 1000        | 0.24.0.07                  | -0.0001           | 0.44.0.10                  | -0.0001           | =
0.30±0.10      | <0.0001           |
|                             | U     | 1900 | -0.24±0.07          | <0.0001           | -0.44±0.10          | <0.0001           | INAIN
NaN-       | 0.0001
NaN∠0.0    |
| East Antarctica             | 0     | 1900 | -0.30±0.10                 | <0.0001 | -0.32±0.10                 | <0.0001 | 0.30±0.10                  | 001                  |

1900 -0.36±0.08 <0.0001 -0.40±0.08 <0.0001 -0.26±0.06 <0.0001

All Antarctica

|                                                              | normalized <del>ne</del>
d                                                              | <del>ormalize</del>                              | CPS CPS                                             |                                                  |                                                      |                                                  |  |
|--------------------------------------------------------------|--------------------------------------------------------------------------------------------|--------------------------------------------------|------------------------------------------------------------|--------------------------------------------------|------------------------------------------------------|--------------------------------------------------|--|
| Geographic regionRegion                                      | Slope
( σ/1000y)sl
<del>ope (σ1000</del>
<del>y-1)</del> | p-
value<del>p</del>
-value | Slope
(°C/1000y) <del>sl</del>
<del>ope</del> | p-
value p-
<del>value</del> | start
date Start
<del>year</del> | end
date End
<del>year</del> |  |
| 1. East Antarctic Plateau
Plateau                         | -0.76±0.28 -
0.76±0.28                                                           | 0 <del>0.000</del>                               | -0.15±0.06 -
<del>0.15±0.06</del>                | 0 <del>0.000</del>                               | 10 <del>10</del>                                     | 1900 <del>1900</del>                             |  |
| 2. Wilkes Land Coast
Coast                                | -0.59±0.48 -
0.59±0.48                                                           | 0.0161
0.016                                  | -0.10±0.07 -
0.10±0.07                           | 0.008 0.0
08                           | 180 <del>180</del>                                   | 1900 <del>1900</del>                             |  |
| 3. Weddell Sea Coast
coast                         | -0.41±0.92 -
0.41±0.91                                                           | 0.3763
0.376                           | $\frac{-0.09\pm0.27}{0.09\pm0.27}$                         | 0.4935 0.
494                          | 1000 <del>1000</del>                                 | 1900 <del>1900</del>                             |  |
| 4. Antarctic Peninsula
Peninsula                          | $\frac{-0.50\pm0.23}{0.50\pm0.23}$                                                         | 00.000                                           | -0.09±0.09 -
0.09±0.09                           | 0.0479 0.
048                          | 00                                                   | 1900 <del>1900</del>                             |  |
| 5. West Antarctic Ice Sheet5.
WAIS                        | -1.32±0.23 -
1.32±0.23                                                    | 00.000                                           | -0.59±0.08-
0.59±0.08                            | 00.000                                           | 0 0                                           | 1900 1900                                 |  |
| 6. Victoria Land Coast 6.
<del>Victoria Ross</del> | -0.89±0.39 -
0.89±0.39                                                    | 0 0.000                                   | $\frac{-0.54\pm0.58}{0.54\pm0.58}$                         | 0.0661 0.
<del>066</del>               | 11401140                                      | 1900 1900                                 |  |
| 7. Dronning Maud Land Coast7.
DML coast                   | 4.98±3.20 4.
98±3.2                                                       | 0.0032
0.003                           | NaN NaN                                             | NaN Na
N                               | 1890 NaN                                      | 1900 NaN                                  |  |
| West Antarctica West Antarctica                              | -0.76±0.16 -
<del>0.76±0.16</del>                                                | 0 0.000                                   | -0.55±0.07 -
<del>0.55±0.55</del>                | 0 0.000                                   | 0 0                                           | 1900 1900                                 |  |
| East AntarcticaEast Antarctica                               | -0.59±0.19 -
0.59±0.19                                                           | 0 0.000                                   | -0.18±0.06 -
0.18±0.06                           | 0 0.000                                   | 10 10                                         | 1900 1900                                 |  |
| All Antarctica All Antarctica                                | -0.76±0.15 -
0.76±0.15                                                           | 0 0.000                                   | -0.38±0.05 -
0.38±0.05                           | 0 0.000                                   | 0 0                                           | 1900 1900                                 |  |

**4.1.3 Trends of the last 100 years in weighted reconstructions**

- 5 Studies based on individual ice cores have identified significant positive trends in the last century (Mulvaney et al., 2012; Steig et al., 2013). Similar to the findings for unweighted composites, significant warming trends in the last 100 years (Table 4) of the weighted anomalies are evident for the Antarctic Peninsula ( $+2.65\sigma 100y^{-1}$  with respect to the 1960-1990 CE normalisation period), Dronning Maud Land coast ( $+2.51\sigma 100y^{-1}$ ) and the West Antarctic Ice Sheet ( $+1.17\sigma 100y^{-1}$ ) regions. The trends in other regions are not significant (Table 4). In temperature units, the NB2014 method gives a scaling about twice as within 30%
- 10 of ECHAM, while CPS is in betweengenerally 50% lower (Table 4). Since 1900\_CE, the reconstructions indicate that the Antarctic Peninsula has been warming by 0.91.14-1.992.87°C 100y-1, West Antarctic Ice Sheet by 0.6646-2.01.32°C 100y-1, and the Dronning Maud Land coast by 0.51-59 to 0.981.33°C 100y-1. The borehole temperature adjustment needed to match the long\_-term trend leads to an over-estimation of the 100-year trend in the WAIS region. Indeed, the same borehole temperature record finds a warming trend of 0.70°C 100y-1 for the past 100 years, and 1.22°C 50y-1 for the past 50 years. This
- 15 example shows that a simple linear scaling cannot be valid for all timescales, and that another approach will be needed to improve quantitative temperature reconstructions for Antarctica.

Despite these uncertainties on absolute scaling, our analyses underline that the sustained warming of the Antarctic Peninsula over the last century stands out as being a robust feature across all methods. Moreover, while the West Antarctic Ice Sheet and the Peninsula regions have now seen reversal of the long-term cooling trend of the past 2000 years, this is not the case for the rest of the continent, where temperatures changes over the last century have not been significant (Figure 4, Table 4).

Table 4: Regression of the last 100-year slope based on the 5-year averages

20

| - |            |       |        |        |                |
|---|------------|-------|--------|--------|----------------|
|   | normalized | ECHAM | NB2014 | Boreho | <del>CPS</del> |
|   |            |       |        | لم     |                |
|   |            |       |        | TC     |                |

| Region                         | Slope                           | <del>p_</del>       | Slope                               | <del>p value</del> | Slope                          | Slope                       | slope                          | <del>p value</del>                      |
|--------------------------------|---------------------------------|---------------------|-------------------------------------|--------------------|--------------------------------|-----------------------------|--------------------------------|-----------------------------------------|
|                                | ( <del>σ</del>
100v - | value               | <del>(°C 100y</del> ⁻• <del>)</del> |                    | (° <del>C 100y</del> ⁻•)       | (°C
100v -1 ) |                                |                                         |
|                                | + <del>)</del>                  |                     |                                     |                    |                                |                             |                                |                                         |
| 1. Plateau                     | <del>-0.85</del>                | <del>0.133</del>    | <del>-0.31</del>                    | <del>0.174</del>   | <del>-1.28</del>               | NaN                         | <del>0.66</del>                | <del>0.024</del>                        |
| 2. Wilkes Coast                | <del>1.08</del>                 | <del>0.115</del>    | <del>0.75</del>                     | <del>0.03</del> 4  | 0.47                           | NaN                         | <del>0.15</del>                | <del>0.586</del>                        |
| 3. Weddell coast               | <del>-1.10</del>                | <del>0.351</del>    | <del>-0.27</del>                    | <del>0.351</del>   | -0.50                          | NaN                         | <del>-0.27</del>               | <del>0.608</del>                        |
| 4. Peninsula                   | <del>2.65</del>                 | <del>0.000</del>    | <del>0.90</del>                     | <del>0.000</del>   | <del>1.99</del>                | NaN                         | <del>1.42</del>                | <del>0.001</del>                        |
| <del>5. WAIS</del>             | <del>1.17</del>                 | <del>0.01</del> 4   | <del>0.66</del>                     | 0.020              | <del>0.97</del>                | <del>2.03</del>             | <del>0.4</del> 4               | <del>0.239</del>                        |
| 6. Victoria Ross               | <del>-0.79</del>                | <del>0.195</del>    | <del>-0.27</del>                    | <del>0.247</del>   | <del>-0.64</del>               | NaN                         | <del>0.05</del>                | <del>0.845</del>                        |
| 7. DML coast                   | 2.51                     | <del>0.012</del>    | <del>0.51</del>                     | <del>0.006</del>   | <del>0.98</del>                | NaN                         | <del>0.74</del>                | <del>0.050</del>                        |
| West Antarctica                | <del>1.50</del>                 | <del>0.001</del>    | <del>0.78</del>                     | <del>0.007</del>   | <del>1.12</del>                | <del>1.85</del>             | <del>0.63</del>                | <del>0.091</del>                        |
| East Antarctica                | 0.04                            | <del>0.924</del>    | <del>0.05</del>                     | <del>0.743</del>   | 0.06                           | <del>NaN</del>              | <del>0.32</del>                | <del>0.082</del>                        |
| All Antarctica                 | <del>0.72</del>                 | <del>0.040</del>    | <del>0.45</del>                     | <del>0.060</del>   | <del>0.99</del>                | 0.21                        | <del>0.40</del>                | <del>0.049</del>                        |
| -                              |                                 |                     | ECH/                                | AM          | NB20                    | )14                  | Bborel                  | nole                             |
| Geographic region       | start                           | end                 | Slope                        | p-value     | Slope                   | p-value              | Slope                          | p-value                          |
| 1 East Antarctic Plateau       | date
1915             | date
2010 | (°C/100y)
-0.49+0.73      | 0 174              | (°C/100y)
-1 28+1 71 | 0.133                       | (°C/100y)
NaN        | NaN                                     |
| 2 Wilkes Land Coast            | 1015                            | 1005                | 1 70+ 0 55                   | 0.034              | 0.47+0.60                      | 0.115                       | NaN                            | NaN                                     |
| 2. Wilkes Land Coast           | 1015                            | 1005                | $0.70 \pm 1.75$                     | 0.351              | $0.47\pm0.00$                  | 0.351                       | NoN                            | NoN                                     |
| A Antaratia Paninsula   | 1015                            | 2010                | $-0.79\pm1.75$                      | 0.331       | -0.30±1.12
1.00±0.75 | 0.331                | NoN                            | NoN                                     |
| 4. Antarcue remnisura   | 1915                            | 2010                | $\frac{2.0/\pm1.12}{1.12.0.02}$     | <0.001   | 1.99±0.75               | <0.001            | Indin                   |                                         |

[revised manuscript text omitted]

---

## Author Response (AR2)

Editor Decision: Publish subject to technical corrections (20 Sep 2017) by Christian Turney

**Comments to the Author:**
Dear Barbara,

Thanks very much. I must apologise for the delay. I've been away and just catching up.

As you know, the PAGES 2k Network is working towards having all data publicly available. Before I accept the manuscript can you resolve the following issues:

1.: The URL link to the data archive for this study returns an error. Please confirm this is functional (prior to publication).

2.: Please allow the editor to review the data that will deposited at NOAA Paleoclimatology

3.: Add Table S1 to the data archive for this study; include it on the NOAA-Paleo landing page.

4.: Double check that all of the "original data URL" links in Table S1 are correct and functional. The ones that end in ".txt" appear to return error messages.

**Authors' answers**

Regarding the point 1: The link will be working after the publication of the paper. The link it is provisional and I have it from NOAA WDC Paleo.

Regarding point 2: I submitted all the data to NOAA and asked them to make the link available for the editors. The data sent to NOAA contains: the data generated in this study as an xls file, the Table S1 with the metadata info and the URL links to the original data sets, a folder containing the T reconstruction generated with the CPS method and used in this study (this was prepared by Raphael Neukom) and finally a folder with the raw data files (using the PAGES2k template) of some of the Dronning Maud Land ice cores that were not previously saved in a public repository).

Regarding point 3: I agree, it was already decided.

Regarding point 4: Now the links in the updated TableS1 xls file are all working on my computer. I checked also with Bruce Bauer at NOAA and he said that he suspects the excel settings are blocking opening URL's directly from a spreadsheet. At the beginning, I had also the same problem, now in the updated Table all seems OK......